# Effect of inorganic material surface chemistry on structures and fracture behaviours of epoxy resin

Tomohiro Miyata [1], Yohei K. Sato[1], Yoshiaki Kawagoe [2] ✉, Keiichi Shirasu [3] ✉, Hsiao-Fang Wang [4], Akemi Kumagai[1], Sora Kinoshita[5], Masashi Mizukami [6], Kaname Yoshida [7], Hsin-Hui Huang[7], Tomonaga Okabe [2,8,9], Katsumi Hagita [10], Teruyasu Mizoguchi [11] & Hiroshi Jinnai [1] ✉

The mechanisms underlying the influence of the surface chemistry of inorganic materials on polymer structures and fracture behaviours near adhesive interfaces are not fully understood. This study demonstrates the first clear and direct evidence that molecular surface segregation and cross-linking of epoxy resin are driven by intermolecular forces at the inorganic surfaces alone, which can be linked directly to adhesive failure mechanisms. We prepare adhesive interfaces between epoxy resin and silicon substrates with varying surface chemistries (OH and H terminations) with a smoothness below 1 nm, which have different adhesive strengths by ~13 %. The epoxy resins within sub-nanometre distance from the surfaces with different chemistries exhibit distinct amine-to-epoxy ratios, cross-linked network structures, and adhesion energies. The OH- and H-terminated interfaces exhibit cohesive failure and interfacial delamination, respectively. The substrate surface chemistry impacts the cross-linked structures of the epoxy resins within several nanometres of the interfaces and the adsorption structures of molecules at the interfaces, which result in different fracture behaviours and adhesive strengths.

Over the past few decades, the development of materials and products that promote energy conservation has become increasingly important for achieving a sustainable society. Within the transportation industry, including aircraft, marine vessels, and automobiles, the substitution of inorganic materials with polymeric materials has been actively promoted to achieve weight reduction and improved energy efficiency[1,2]. For instance, epoxy resin adhesives have replaced metal rivets and bolts in joining applications. Additionally, fibre-reinforced plastic, a composite comprising epoxy resin as the matrix and glass/carbon fibres as fillers, has been utilized in vehicle body panels for several decades. Epoxy resin is a crucial polymeric material in industrial applications owing to its exceptional thermal and adhesive properties and excellent workability[1–3].

Although epoxy-resin-based materials are energy-efficient alternatives to metals, they have a higher risk of mechanical failure. Major failure modes include delamination at the epoxy resin–inorganic material interfaces, which has long been recognized as a critical limitation of such composites[4–7]. Therefore, gaining a comprehensive understanding of the adhesion mechanisms and delamination/fracture behaviours occurring at the adhesive interfaces between epoxy resins and inorganic materials is desirable for effectively controlling the properties of epoxy resin-based adhesives and composites and improving their performance in a wide range of practical applications.

Adhesion mechanisms typically comprise two factors[8]: (i) chemical interactions, which involve chemical bonds such as covalent and

hydrogen bonds, as well as van der Waals interactions between the materials in contact; and (ii) mechanical interactions that occur through the interlocking of two materials. Additionally, the interfacial regions play a critical role in the fracture behaviour as these areas can have weaker mechanical performance than the bulk materials[9,10]. The relationship between epoxy resin structures and their fracture behaviour near adhesive interfaces is of considerable interest in the development of adhesive systems resistant to interfacial fracture and delamination[11].

Several studies have investigated the adhesive interfaces between epoxy resins and inorganic materials, aiming to elucidate the characteristics of the interfacial regions. X-ray reflectivity and scanning force microscopy-based force modulation microscopy have revealed that epoxy resins near an adhesive interface demonstrate confinement and higher stiffness than the bulk regions[12,13]. The thicknesses of the interfacial regions vary from tens of nanometres to 1 μm, depending on the specific materials and measurement techniques used. Additionally, it has been observed that the amine-to-epoxy (stoichiometric) ratio differs in the interfacial regions compared to the bulk[13,14].

The surface chemistry of inorganic substrates is recognized as a significant factor affecting the adhesive strengths of epoxy resins[2,11,15,16]. Furthermore, molecular dynamics (MD) simulations of curing reactions have shown that the surface chemistry can alter the cross-linked network structures of epoxy resin near adhesive interfaces[17]. However, the relationship between the surface chemistry and the cross-linked structures of epoxy resins near experimental adhesive interfaces remains unexplored. To obtain insights into this relationship, analytical techniques with a spatial resolution higher than the sizes of epoxy-resin monomers are necessary because the curing of epoxy resins involves interactions between epoxy resin molecules and inorganic surfaces.

Transmission electron microscopy (TEM) and scanning TEM (STEM) are powerful tools for observing atomic and molecular structures within materials at high resolution. Furthermore, STEM-based electron energy-loss spectroscopy (STEM-EELS) enables the examination of the spatial distributions of elements and chemical bonding states in materials. In 2022, STEM-EELS was used to investigate the rough interfaces between epoxy resins and aluminium substrates, but the spatial resolution (<5 nm) was insufficient to elucidate the effects of the surface chemistry on the cross-linked structures of epoxy resins near the adhesive interfaces[18]. Furthermore, the use of rough aluminium surfaces resulted in both chemical and mechanical interactions contributing to adhesion. This made it impossible to determine the individual effects of surface chemistry and surface roughness on the adhesion mechanisms. To comprehensively understand the effects of surface chemistry in isolation, it is necessary to conduct sub-nanometre-resolution observations of nearly atomically flat interfaces with various surface chemistries. Moreover, it is crucial to identify fracture paths with high resolution to establish a correlation between the epoxy resin structures and the fracture mechanisms near the adhesive interfaces.

In this study, we prepared planar adhesive interfaces between epoxy resin and silicon substrates with two different surface chemistries: hydroxyl and hydrogen terminations. To investigate these interfaces, we employed three techniques with molecular-scale resolutions: (i) STEM-EELS measurements, (ii) curing MD simulations, and (iii) fracture TEM observations. By integrating these techniques, we were able to provide new insights into the chemical compositions, cross-linked structures, and fracture surfaces near the adhesive interfaces. Finally, we explored the adhesion and fracture mechanisms that influence adhesive strength, which exclusively stem from chemical interactions and nanoscale molecular structures near the adhesive interfaces, thereby excluding mechanical interactions.

## Results and discussion

### Adhesive strengths between epoxy resin and Si substrates

We prepared adhesive interfaces between epoxy resin and Si substrates with OH- and H-terminated surfaces. Hereafter, we refer to these interfaces as the OH- and H-terminated interfaces. The epoxy resin used in this study was a mixture of bisphenol A diglycidyl ether (DGEBA) and 4,4-diaminodicyclohexylmethane (PACM). The chemical structures of DGEBA and PACM are depicted in Fig. 1. Lap-shear tests were conducted to measure the adhesive strengths of the OH- and H-terminated interfaces, yielding values of 34.8 ± 1.2 and 30.8 ± 1.8 MPa, respectively. These results indicate that the OH-terminated interface exhibits a higher average adhesive strength than the H-terminated interface. The surfaces of the Si substrates were characterized by a high level of flatness, with an arithmetic mean roughness of <0.5 nm determined using atomic force microscopy (AFM). Therefore, the adhesion between the epoxy resin and Si substrates is majorly governed by chemical interactions rather than mechanical interactions resulting from surface roughness. Essentially, the variation in adhesive strengths mainly stems from discrepancies in the surface chemistry of the Si substrates. The subsequent sections describe nanoscopic analyses of the adhesive interfaces to clarify the impact of the surface chemistry of the Si substrate on the cross-linked structures and fracture mechanisms of the epoxy resin close to these interfaces.

### Electron microscopy images of the adhesive interfaces

Figure 2 presents annular dark-field STEM (ADF-STEM) images of the OH- and H-terminated interfaces observed along the <110> direction of the Si substrates, which correspond to single Si crystals. The columns of Si atoms are observed as bright spots (upper portions of Fig. 2a, b). Specifically, the OH- and H-terminated interfaces align with the {111} crystal planes. The epoxy resin layers in the lower regions of the images appear darker due to the presence of lighter elements (H, C, N, and O). At both the OH- and H-terminated interfaces, silicon oxide ($SiO_2$) layers with ~2 nm thickness were observed between the Si crystal and epoxy resin. The native oxide layers on the Si substrates were not removed during the surface treatment with dilute hydrofluoric (HF) acid (1.5 wt.%) and water-plasma treatments used to achieve the OH and H terminations. This was further supported by the thickness measurements of the oxide layer using ellipsometry (Supplementary Table 1). The characterization of the Si surfaces confirmed the presence of OH and H terminations achieved through the HF acid and water-plasma treatments, respectively. This was determined through water contact angle measurements and ATR-FTIR spectra, as shown in the Supplementary Information (Supplementary Figs. 1 and 3). Additionally, ellipsometry measurements (Supplementary Table 1) and ADF-STEM cross-sectional observations (Fig. 2) revealed that the natural oxide layer of the Si substrate remained intact after the HF acid treatment due to the significantly diluted (1.5 wt.%) nature of the HF acid solution.

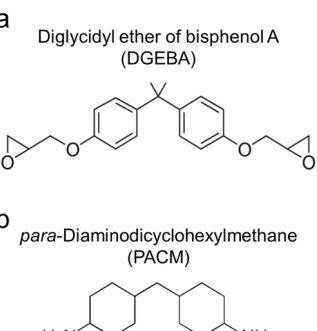

**Fig. 1 | Molecular structures of the components of the epoxy resin. a** DGEBA and **b** PACM.

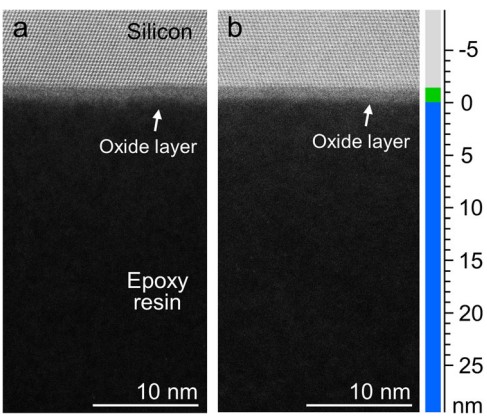

**Fig. 2 | Cross-sectional annular dark-field scanning transmission electron microscopy (ADF-STEM) images of the adhesive interfaces. a** OH-terminated and **b** H-terminated interfaces. Silicon oxide (SiO$_2$) layers are observed in both images. Source data are provided as Source Data files.

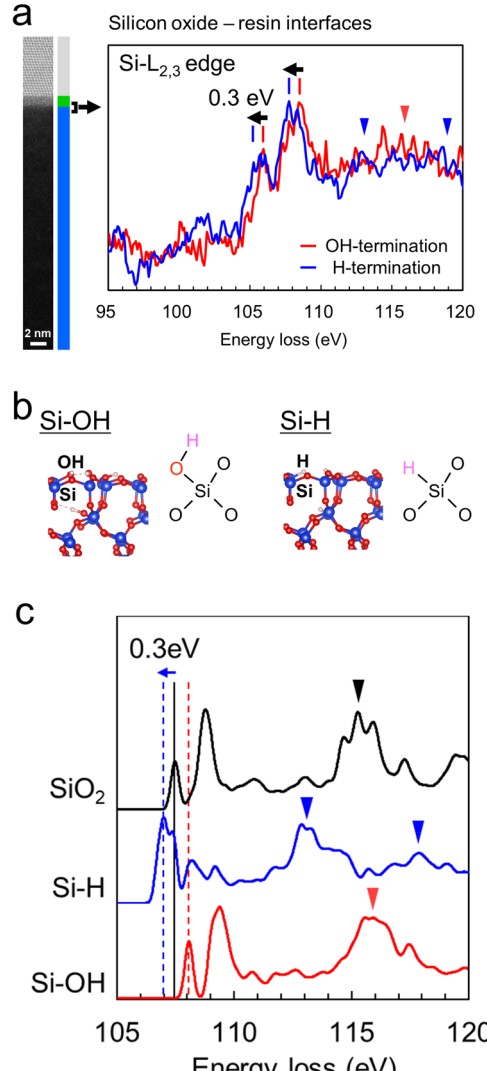

**Fig. 3 | Si-L$_{2,3}$ edge electron energy-loss near-edge structure (ELNES) spectra of SiO$_2$ surfaces. a** Experimental Si-L$_{2,3}$ edge spectra measured at the interfaces between the epoxy resin and OH- and H-terminated oxide layers with a width of 1 nm. **b** Surface models of OH- and H-terminated (0001) surfaces of α-quartz used for simulating Si-L$_{2,3}$ edge spectra. **c** Simulated Si-L$_{2,3}$ edge spectra from the Si atoms at the outermost surface of the models.

Consequently, the oxide layers on the Si substrates with different surface chemistries exhibited similar thicknesses. This facilitated a comparative analysis focused on the influence of surface chemistry on the interphase structure and fracture behaviour, with minimal consideration of surface morphology.

### Surface chemistry of the Si substrates with epoxy resin

The surface chemistry of the Si substrates after pasting and curing the epoxy resin was analysed using STEM-EELS. Figure 3 shows the electron energy-loss near-edge structure (ELNES) spectra of the Si-L$_{2,3}$ absorption edges at the OH- and H-terminated interfaces. These spectra were precisely measured at the interfaces between the epoxy resin and 1 nm-wide oxide layers, as shown on the left side of Fig. 3a. The red and blue spectra correspond to the OH- and H-terminated interfaces, respectively. The spectrum of the OH-terminated interface resembles that of SiO$_2$, with peaks observed at 106, 108, and 115 eV (as indicated by the vertical red lines and triangles in Fig. 3a). These characteristic peak positions indicate a chemical bonding state of Si atoms at the OH-terminated interface similar to SiO$_2$, with the Si atoms adopting a tetrahedral arrangement, surrounded by four O atoms[19–22]. In the spectrum of the H-terminated interface, the peaks at 106 and 108 eV are shifted towards lower energy by 0.3 eV compared to the spectrum of the OH-terminated interface.

To investigate the origin of the peak shifts, surface models were prepared with Si-OH and Si-H bonds, representing the OH and H terminations, respectively. An α-quartz (0001) dense surface was used as the model for the SiO$_2$ layers (Fig. 3b). Figure 3c displays the simulated Si-L$_{2,3}$ edge spectra for the Si atoms on the uppermost surface of these surface models. The black spectrum corresponds to the Si atoms on the surface of the α-quartz (0001) dense surface, with no groups other than Si-O-Si bonds. The red and blue spectra were simulated for the Si atoms in Si-OH and Si-H bonds, respectively. These simulated spectra reveal that the first peak in the spectrum of the Si atoms with a Si-H bond (H termination) exhibited a chemical shift towards the lower-energy side by 0.3 eV compared to that of the unmodified SiO$_2$ model. Conversely, the first peak shifted towards the higher-energy side by 0.5 eV for the Si atoms with a Si-OH bond (OH termination). These chemical shifts in the Si-H and Si-OH bonds resulted from the increased and decreased charges of the Si atoms, respectively, compared to the Si-O-Si termination.

In the STEM-EELS experiments, simultaneous measurements were performed on Si atoms inside the oxide layers and on their surfaces. Consequently, the spectra of the Si-H/Si-OH bonds on the surfaces overlapped with those of the oxide layers (SiO$_2$ bulk region). This explains why the experimentally observed spectra resemble that of

SiO$_2$ (Fig. 3a). Moreover, the chemical shift towards the lower-energy side in the experimental spectrum suggests the presence of numerous Si-H bonds that remain at the H-terminated interface, even after being covered with epoxy resin. Furthermore, in the spectrum of the H-terminated interface, the peak at 115 eV was not clearly observed, while additional peaks are observed at 112 and 118 eV. This change in the peak positions suggests that the Si atoms at the H-terminated interface exhibit distinct environments compared to the SiO$_4$ tetrahedral arrangement.

To explore other potential chemical bonds at the interfaces between the SiO$_2$ layer and the epoxy resin (DGEBA-PACM system), additional surface models were prepared, incorporating Si-O-phenyl, Si-O-C, Si-N, and Si-C bonds (Supplementary Fig. 6). Subsequently, Si-L$_{2,3}$ edge ELNES simulations were conducted based on these models. As illustrated in Supplementary Fig. 7, only the spectrum of the Si-C bond displays a chemical shift towards the lower-energy side when compared to the unmodified SiO$_2$. Previous reports suggest that Si-H bonds can react with epoxy groups (albeit under UV irradiation), indicating the likelihood of Si-H bonds reacting with DGEBA molecules

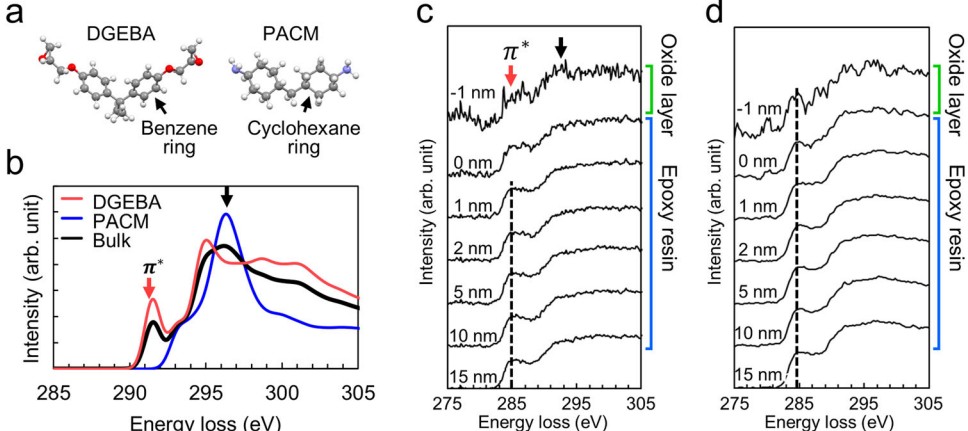

**Fig. 4 | C–K edge spectra of the epoxy resin in the DGEBA-PACM system.**
**a** Molecular structures of DGEBA and PACM used for the electron ELNES simulation.
**b** Simulated C–K edge spectra for DGEBA, PACM, and the bulk of the epoxy resin
(stoichiometric ratio of 1:1). The energy loss of the simulated spectra is not directly

compared to that of the experimental spectra. Experimental C–K edge spectra
measured at various positions near the **c** OH-terminated and **d** H-terminated
interfaces in 1 nm steps. The measured positions are indicated relative to the
interface.

at the H-terminated interface. However, in such instances, a stable
structure is formed by creating Si-O-C bonds rather than Si-C bonds[23].
Consequently, the observed chemical shift in the Si-$L_{2,3}$ edge spectrum
towards the lower-energy side originates from the presence of Si-H
bonds (H termination).

**Compositional heterogeneity near the interfaces**
The compositional ratio of DGEBA to PACM near the OH-
and H-terminated interfaces was investigated using STEM-EELS
(Supplementary Fig. 10). As depicted in Fig. 4a, b, the presence of
aromatic rings (benzene rings) in the DGEBA molecular structure
results in the appearance of a π* peak at the C–K edge in the EEL
spectrum. The π* peak originates from the electron transition from
the 1 s to the π* orbitals in C atoms. In contrast, the spectrum of
PACM, without π bonds in its molecular structure, does not exhibit
π* peaks at the C–K edge. These distinct characteristics facilitated
the identification of compositional heterogeneity within the epoxy
resin. Furthermore, the energy loss in the simulated spectra differs
from that observed in the experimental spectra by ~6 eV, attributed
to calculation errors.

Figure 4c, d present the C-K edge spectra measured near the OH-
and H-terminated interfaces, respectively. At every measured position,
an π* peak at 285 eV and a shoulder-like σ* structure at 292 eV
(representing the transition from 1 s to σ* orbitals) were observed. In
the case of the OH-terminated interface, a decrease in the intensity of
the π* peak (red arrow) was observed around the surface of the oxide
layer (−1 - 0 nm from the interface) compared to the bulk region of the
epoxy resin (1–15 nm from the interface). This decrease in intensity
signifies an increased ratio of PACM to DGEBA at the OH-terminated
interface. Furthermore, the increased intensity at 292 eV (black arrow)
supports the condensation of PACM at the OH-terminated interface.
In contrast, the C–K edge spectra at the H-terminated interface exhi-
bits a clear π* peak with comparable intensity to that observed in
the bulk region, suggesting minimal compositional changes at the
H-terminated interface.

The profiles of the stoichiometric ratio (amine-to-epoxy ratio)
were evaluated by dividing the π* peak intensity by the σ* peak
intensity in the C–K edge spectrum, as described in Supplementary
Note 10 in the Supplementary Information. These profiles in Fig. 5 (red
circles) show that the stoichiometric ratio at the OH-terminated
interface (~1.3) is larger than that in the bulk (1.0). In contrast, the
epoxy resin close to the H-terminated interface shows a stoichiometric
ratio of ~0.7, which is smaller than that of the bulk region.

To verify the stoichiometry changes near the interfaces, compo-
sitional analysis of the epoxy resin was conducted using N–K-edge
ELNES of the OH- and H-terminated interfaces. Since only PACM con-
tains N atoms, the N–K edge spectra were attributed to PACM mole-
cules. These spectra are presented in Supplementary Fig. 8. Although
the N–K edge spectra exhibit weak signal intensities, making it chal-
lenging to analyse their spectral features, the relative amount of N
atoms (representative of PACM) near the interfaces was determined by
integrating the intensity of each N–K edge spectrum within the energy
range of 400–420 eV. Furthermore, the integrated intensities of the
C–K edge spectra within the range of 280–300 eV are approximately
proportional to the quantity of all molecules, including DGEBA and
PACM. Therefore, the integrated N–K edge intensity was normalized
by the corresponding integrated C–K edge intensity, referred to as the
"N–K/C–K intensity," to estimate the relative ratio of PACM at each
position from the interfaces. The blue squares in Fig. 5a, b depict the
stoichiometric ratios calculated from the N–K/C–K intensities plotted
against the distance from the OH- and H-terminated interfaces,
respectively. In the case of the OH-terminated interface, the stoichio-
metric ratio of PACM is higher near the oxide-layer interface compared
to the bulk region of the epoxy resin. Conversely, a gradual decrease in
the stoichiometric ratio is observed for the H-terminated interface
from the bulk toward interfacial regions. These profiles are consistent
with those obtained from the C–K edge analysis.

The attenuated total reflection–Fourier transform infrared spec-
troscopy (ATR-FTIR) spectrum of the OH-terminated interface shows a
higher peak intensity for the $CH_2$ vibrations compared to the
H-terminated surface (Supplementary Fig. 4), supporting the observed
condensation of PACM in that region. This is consistent with the higher
amount of $CH_2$ groups in PACM compared to DGEBA, as discussed in
Supplementary Note 6 of the Supplementary Information. Further-
more, the detection of PACM condensation by ATR-FTIR suggests that
it is a prevalent feature across the entire OH-terminated interface.

**Curing simulation of bulk systems of epoxy resins**
The variations in the stoichiometric ratios of DGEBA to PACM should
result in different cross-linked structures. To evaluate the cross-linked
structures of the epoxy resin near the interfaces, we conducted curing
MD simulations that consider the formation of cross-linked structures
through exothermal reactions[24]. Fig. 6a shows an example of a periodic
cell model used to investigate the cross-linked structures formed in
bulk systems with different stoichiometric ratios. The simulation
considered the primary and secondary amine-epoxy reactions

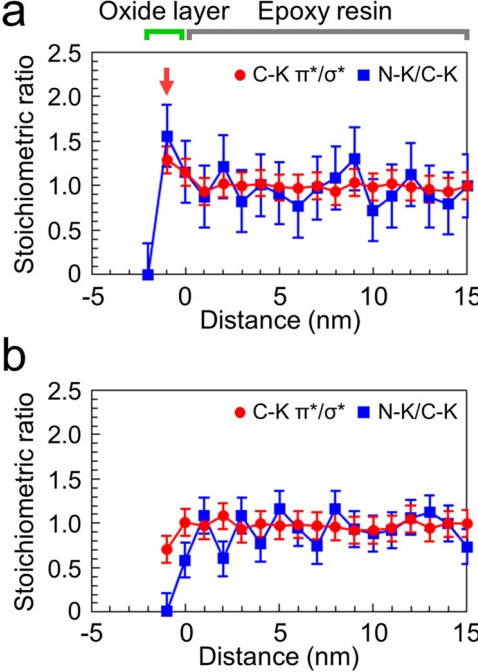

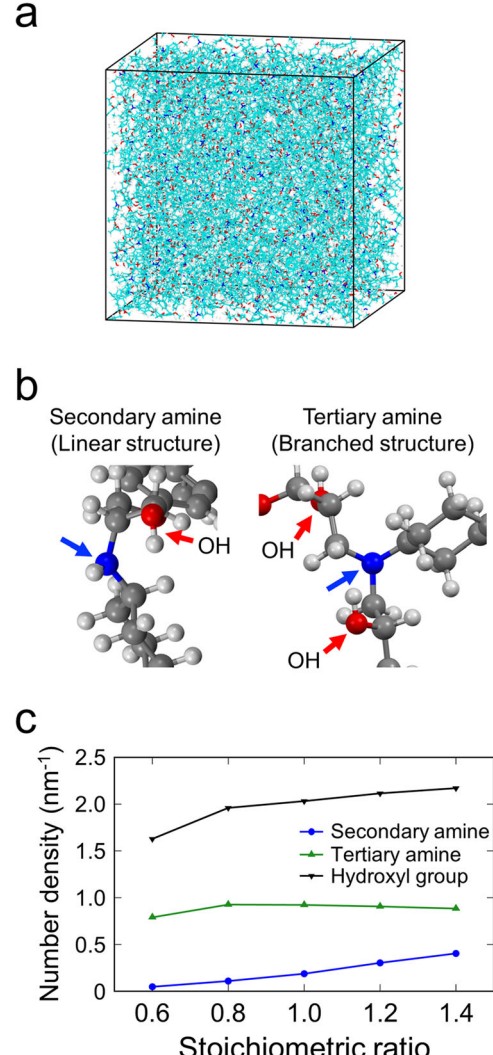

**Fig. 5 | Profiles of the stoichiometric ratio (amine-to-epoxy ratio) measured at different distances from the adhesive interfaces. a** OH-terminated and **b** H-terminated interfaces. The red circles represent the stoichiometric ratio estimated from the intensity of the π* peak divided by that of the corresponding σ* peak of the C–K edge spectrum (C–K π*/σ*). The blue squares indicate the stoichiometric ratio estimated from the integrated intensity of the N–K edge spectrum divided by that of the corresponding C–K edge spectrum (N–K/C–K). The error bars for the profiles estimated from the C–K π*/σ* intensities were defined as the errors of the curve fitting to the C–K edge spectra. The error bars for the profiles estimated from the N–K/C–K intensities were defined as the deviations from the average values in the regions within 5–15 nm from the interface, assuming that the stoichiometric ratio would be uniform in the region.

**Fig. 6 | Curing molecular dynamic simulations of bulk systems of DGEBA-PACM epoxy resins with various stoichiometric ratios. a** Model of the bulk system used for the simulations. **b** Schematics of the secondary and tertiary amine structures. Hydroxyl groups are generated by the reactions to form secondary and tertiary amines. **c** Number densities of the secondary and tertiary amines and hydroxyl groups as a function of the stoichiometric ratio.

(Supplementary Fig. 5), forming linear secondary and branched tertiary amines and hydroxyl groups (Fig. 6b). Figure 6c shows that the number density of the tertiary amines (branched structures) is almost independent of the stoichiometric ratio, while those of the secondary amines (linear structures) and hydroxyl groups increase with increasing stoichiometric ratio.

## Curing simulation on epoxy resin-silica interfaces

Subsequently, curing MD simulations of interfacial models were performed to explore the cross-linked and adhesion structures close to the OH- and H-terminated interfaces in detail. In the simulation, the (10$\bar{1}$0) surfaces of α-quartz were terminated with OH and H as models of the oxidized Si layers, as shown in Fig. 7a. The uncured epoxy resin, consisting of DGEBA and PACM molecules, was confined between the OH- and H-terminated surfaces. The average stoichiometric ratios of the uncured epoxy resins within the OH- and H-terminated systems were set to reflect the interfacial compositions measured by STEM-EELS. Then, curing MD simulations were performed on these systems. Note that the simulations did not consider chemical reactions between the epoxy resin molecules and the substrates.

The stoichiometric ratios of the cured epoxy resins within 1 nm of the OH- and H-terminated interfaces (grey region in Supplementary Fig. 11) are ~1.5 and 1.0, respectively, which are similar to the experimentally measured ratios. Figure 7b depicts the number density distributions of the reacted amino groups (secondary and tertiary amines) in the epoxy resins near the OH- and H-terminated interfaces. Within 1 nm of the interface (grey regions), the number densities of

tertiary amines (branched structures) are almost equal for both the OH- and H-terminated interfaces, whereas the OH-terminated interface, with a higher stoichiometric ratio, has more secondary amines (linear structures) than the H-terminated interface. This tendency is in good agreement with that of bulk systems, as shown in Fig. 6c. In other words, the OH-terminated interface has a higher average molecular weight between the cross-links than the H-terminated interface. The presence of distinct cross-linked structures at the interfaces could result in different mechanical properties[25].

The red lines in Fig. 7b show the distributions of the hydroxyl groups, originating from the reactions between epoxy and amino groups, as shown in Supplementary Fig. 5. The OH-terminated system exhibits a higher average density of the hydroxyl groups in the epoxy resin compared to the H-terminated system. This originates from the higher stoichiometric ratio in the OH-terminated system, as observed for the bulk system (Fig. 6c). Furthermore, the adsorption layers of hydroxyl groups exhibit a higher density near the OH-terminated interface compared to that near the H-terminated interface (denoted by black triangles). This difference arises from the formation of

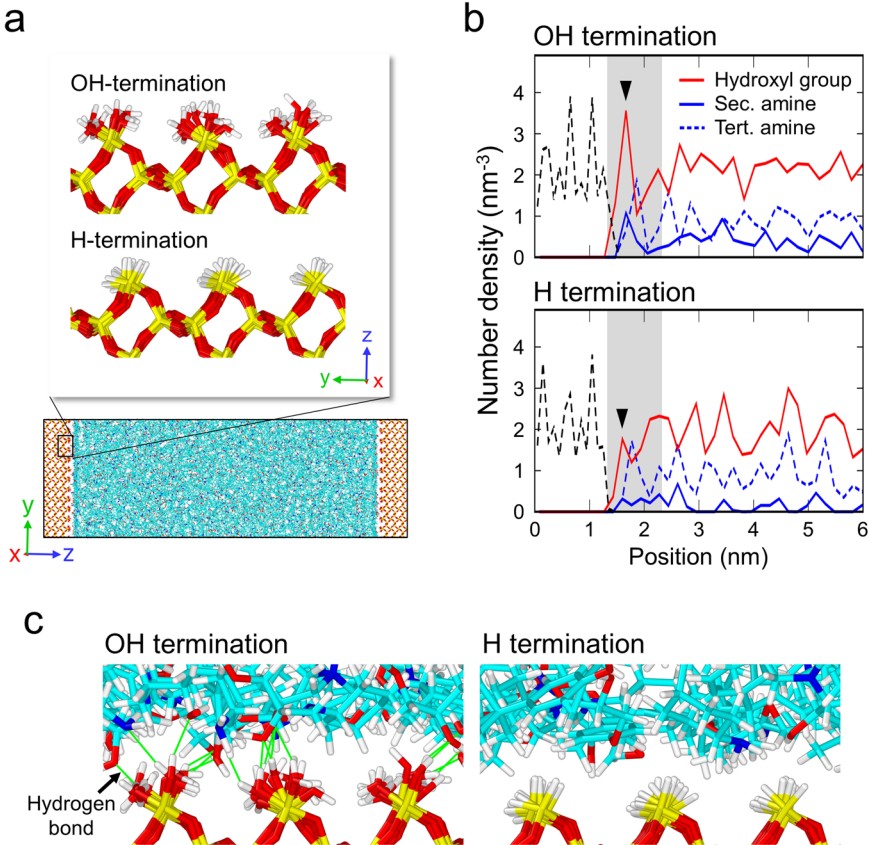

**Fig. 7 | Epoxy resin structures cured at the OH- and H-terminated interfaces.**
**a** Simulation model before curing and the surface structures of the OH- and
H-terminated SiO$_2$ ($\alpha$-quartz) used for the curing molecular dynamics simulations.
The top faces of the SiO$_2$ crystals are terminated with hydroxyl groups and
hydrogen atoms, respectively. **b** Number densities of the secondary and tertiary
amines and hydroxyl groups in the epoxy resins near the OH- and H-terminated
interfaces. The number densities of Si and O atoms in the substrates are shown by
black dotted lines to clarify the positions of the interfaces. The regions within 1 nm
of the interfaces are represented by grey. **c** Schematics of the OH- and H-terminated
interfaces.

hydrogen bonds between the hydroxyl groups on the silica surface
(silanol groups) and the hydroxyl groups generated within the
epoxy resin.

Figure 7c illustrates the simulated atomic structure of the OH- and
H-terminated interfaces. The hydrogen bonds (hydrogen-oxygen dis-
tance: <0.25 nm) between the silanol groups of the substrate and
hydroxyl groups of the epoxy resin are indicated by green lines. The
illustration indicates that numerous hydrogen bonds are formed at the
OH-terminated interface, while there are very few hydrogen bonds at
the H-terminated interface. Consequently, the OH-terminated inter-
face with a PACM-rich composition generates many hydroxyl groups in
the epoxy resin and preferentially forms hydrogen bonds to stabilize
the adsorption structure.

The cross-linked structures exhibit stronger adhesion to the OH-
terminated surfaces compared to the H-terminated surfaces due to
hydrogen bonding. To quantitatively analyse the different interactions
at the OH- and H-terminated interfaces, the adhesion energies of the
epoxy resins on these surfaces were evaluated. The adhesion energy
($E_{ad}$) between the epoxy resin and the surfaces is defined as follows[26]:

$$E_{ad} = (E_{epo} + E_{sub}) - E_{epo-sub} \qquad (1)$$

Here, $E_{epo-sub}$ is the total energy of the interfacial model, and ($E_{epo}$ +
$E_{sub}$) is the total energy when the epoxy resin and SiO$_2$ substrates are
separated. The adhesion strength is quantified by the magnitude of
$E_{ad}$, where a higher value indicates stronger adhesion.

The $E_{ad}$ values estimated for the OH- and H-terminated interfaces
from the curing MD simulations were 0.223 and 0.147 J m$^{-2}$,

respectively. These values indicate that the OH-terminated interface is
energetically more stable and exhibits stronger surface adsorption.
Although the magnitude of $E_{ad}$ depends on the potential parameters
utilized in the curing MD simulation, the relationship between the $E_{ad}$
values of the OH- and H-terminated interfaces is consistent. These
findings are supported by the contact angle analysis (Supplementary
Note 3 and Supplementary Fig. 2). The contact angles of DGEBA,
PACM, and the DGEBA-PACM mixtures on the OH-terminated surface
are smaller than those on the H-terminated surface. This suggests that
the OH-terminated surface energetically stabilizes both DGEBA and
PACM molecules to a greater extent than the H-terminated surface,
resulting in stronger molecular adsorption on the OH-terminated
surface. The higher $E_{ad}$ at the OH-terminated interface indicated by the
MD simulation and contact-angle results is consistent with the results
of a lap-shear test (Supplementary Fig. 12), where the OH-terminated
interface demonstrated a stronger adhesive strength than the
H-terminated interface. Consequently, the differences in cross-linked
structures and $E_{ad}$ at the OH- and H-terminated interfaces, which are
dependent on the surface chemistry, are likely to influence the
mechanical properties near the interfaces, particularly the fracture
behaviour.

**Fracture surfaces observed by TEM**
To elucidate the fracture mechanisms near the OH- and H-terminated
interfaces, cross-sectional TEM observations of fracture surfaces were
conducted. Cross-sectional TEM specimens of the OH- and
H-terminated interfaces, with a thickness of ~300 nm, were prepared
using a focused ion beam (FIB). Subsequently, tensile loads were

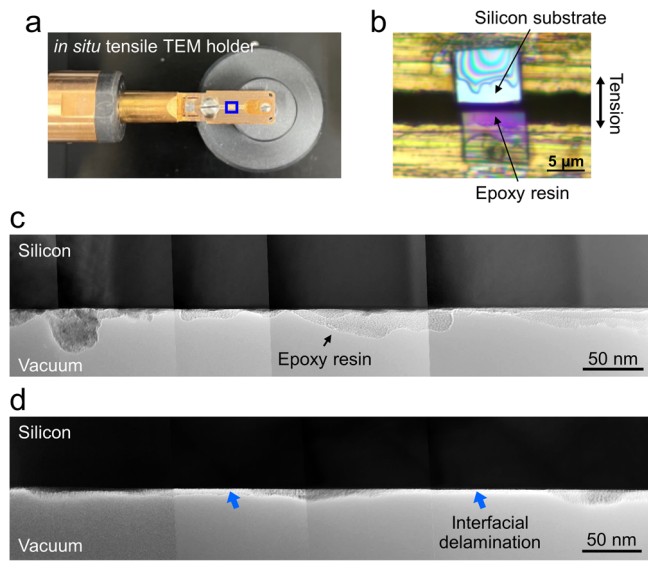

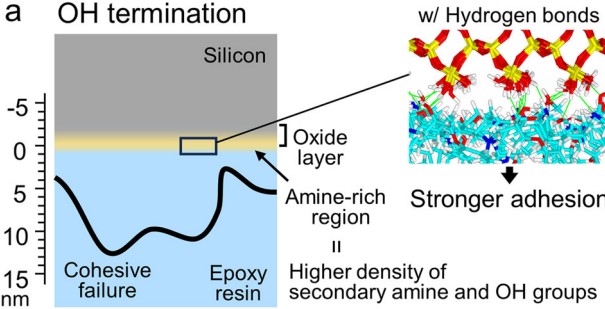

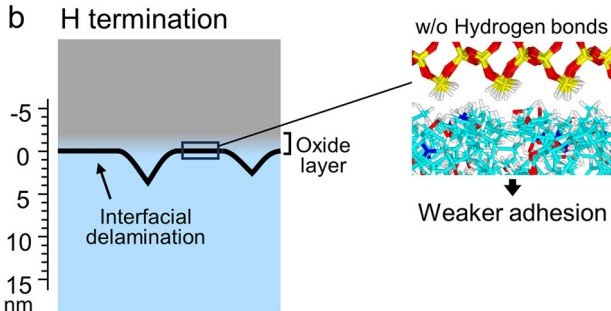

**Fig. 8 | Transmission electron microscopy (TEM) observation of the fracture surfaces near the OH- and H-terminated interfaces. a** Photograph of the tensile specimen holder with a tensile cartridge. **b** Magnified image of the region highlighted by the blue square in (**a**), showing an interface specimen bridged over the slit of the tensile cartridge. A tensile stress was applied to the interface between the epoxy resin and Si substrate to induce fracture as the slit was opened. Cross-sectional TEM images of the fracture surfaces near the (**c**) OH-terminated and (**d**) H-terminated interfaces.

**Fig. 9 | Schematics of the interfacial structures and fracture paths near the adhesive interfaces. (a)** OH-terminated and (**b**) H-terminated interfaces.

applied perpendicular to the interfaces using a tensile TEM holder to induce fracture, as depicted in Fig. 8a, b[27–29]. Although the loading conditions applied to the specimens of the lap-shear test were different from those of the tensile fracture tests with the TEM holder, we presumed that the strength relationships between the OH- and H-terminated interfaces were similar for both tensile and shear deformation since interfaces with weaker bonds are more likely to break at lower loads.

Figure 8c, d present the fracture surfaces of the OH- and H-terminated interfaces, respectively. In these cross-sectional TEM images, residual epoxy resin is observed on the top surfaces of the Si substrates at both the OH- and H-terminated interfaces. The average thicknesses of the residual epoxy resin layers are ~11 and ~6 nm for the OH- and H-terminated interfaces, respectively. Additionally, while the fracture surface of the OH-terminated interface is entirely located within the epoxy region (cohesive failure), half of the fracture surface of the H-terminated interface is at the interface itself (interfacial delamination).

The distinct fracture surfaces and mechanisms near the OH- and H-terminated interfaces are attributed to differences in the surface chemistry, i.e., different cross-linked structures and $E_{ad}$ values. As depicted in Fig. 9, the OH-terminated interface exhibits higher $E_{ad}$, leading to fractures occurring within the epoxy resin (cohesive failure) without interfacial delamination. The non-linear fracture path is attributed to the heterogeneity in the cross-linked structure of the epoxy resin[30]. In contrast, the interfacial delamination of the H-terminated interface is attributed to the lower adhesion energy at the interface. However, residual epoxy resin was observed on the surface of the Si substrate. H termination on Si substrates has been reported to be replaced by OH termination over time when the material is exposed to air[31,32]. In this case, the areas with OH termination would exhibit similar fracture behaviours to the OH-terminated interface, namely cohesive failure, even at the H-terminated interface.

The exclusive condensation of PACM at the OH-terminated interface indicates distinct cross-linked structures of the epoxy

resins near the OH- and H-terminated interfaces, even at the nanoscale. As cracks propagated within a single nanometre from the adhesive interfaces, the dissimilarity in the cross-linked structures of the epoxy resin near the interfaces influences the energy required for crack propagation. The distinct fracture mechanisms and cross-linked structures of the epoxy resins near the OH- and H-terminated interfaces led to different adhesive strengths, as measured by the lap shear test.

This study firstly examined the effects of the surface chemistry (OH and H termination) of Si substrates on the composition, cross-linked structure, and fracture mechanisms of epoxy resin near adhesive interfaces. Lap shear tests were conducted to quantify the adhesive strength, showing that the OH-terminated interface exhibited ~13% higher adhesive strength than the H-terminated interface. To understand the underlying reasons for this discrepancy in adhesive strength, high-resolution TEM/STEM and curing MD simulations were employed for detailed analysis.

The Si-$L_{2,3}$ edge ELNES spectra provided insights into the H termination remaining even after the resin was applied and cured on the surface. In contrast, the C-K and N-K edge ELNES spectra indicated that the PACM curing agent condensed (resulting in a higher stoichiometric ratio than that of the bulk) within a 1–2 nm range at the OH-terminated interface, whereas the opposite phenomenon was observed at the H-terminated interface (smaller stoichiometric ratio than that of the bulk). These findings were corroborated by ATR-FTIR measurements. The Si substrates have a sufficiently flat surface with a roughness of ~0.5 nm, and the oxide layers have comparable thicknesses at both the OH- and H-terminated interfaces. Given these conditions, we conclude that the disparity in surface chemistry solely accounts for the variations in chemical composition near the interfaces.

Curing MD simulations further indicated that the compositional differences in the epoxy derived from the surface chemistry of the Si substrate influenced the cross-linked structures near the interfaces. Additionally, the adhesion energy at the OH-terminated interface was higher (indicating greater stability) than that of the H-terminated interface due to the adsorption of the epoxy resin on the OH-terminated surface via hydrogen bonds.

TEM observations of the fractured surfaces revealed distinct fracture behaviours between the OH-terminated surface and H-terminated interfaces. The OH-terminated interface exhibited cohesive failure due to its higher adhesion energy, while the H-terminated interface showed partial interfacial delamination due to its lower adhesion energy.

These findings clearly demonstrated the significant influence of the intermolecular interactions at adhesive interfaces the surface chemistry of inorganic substrates on both the molecular segregation and cross-linked structures of epoxy resin interfacial interactions between the epoxy resin and substrates and the epoxy resin cross-linked structures near the interfaces at a single-nanometre scale. The structural changes in the epoxy resin due to the surface chemistry near the interfaces are directly linked to have implications for the interfacial adhesive strengths and the propagation path of cracks near the interfaces. The acquired knowledge of the chemical interactions at the adhesive interfaces contributes to a comprehensive understanding of the interfacial adhesion and fracture mechanisms between polymers and inorganic materials.

## Methods

### Surface modification of Si substrates

Si single crystals with a surface orientation of (111) and a purity exceeding 99.999% were obtained from Crystal Base Co., Ltd., Japan. The Si substrates underwent surface modifications to create H- and OH-terminations through the following procedures. Initially, all organic residues on the Si surface were eliminated by immersing the Si substrate in a piranha solution (3:1 ratio of sulfuric acid to hydrogen peroxide). Subsequently, the hydroxyl groups on the Si surface were eliminated by immersing the Si substrate in a dilute solution of HF acid (1.5 wt.%) for 30 min to generate H-terminated Si surfaces. Additionally, some of the H-terminated surfaces underwent treatment with water vapour plasma for 20 min to introduce hydroxyl groups[33].

### Measurement of contact angles on Si substrates

The contact angles of water, DGEBA (jER YL6810, Mitsubishi Chemical Corporation, purity: 100 %), PACM (Tokyo Chemical Industry Co., Ltd., purity: >97.0 %), and a DGEBA−PACM mixture with a stoichiometric ratio of 1:1 (pre-cured epoxy resin) on OH- and H-terminated Si substrates were measured. Water contact angles were measured at room temperature. In contrast, the contact angles of DGEBA, PACM, and the DGEBA-PACM mixture were measured at 60 °C (above the melting temperatures of DGEBA (40–44 °C) and the glass transition temperature of PACM (30–60 °C) determined by differential scanning calorimetry). The contact angle $\theta$ values of the droplets on the Si substrates were determined using the $\theta/2$ method and the following equation:

$$\theta = 2 \cdot \arctan\left(\frac{h}{r}\right) \qquad (2)$$

Where $h$ and $r$ denote the height and radius of the droplet on the substrate, respectively.

Water contact angle, ATR-FTIR, and ellipsometry measurements were conducted within 30 min of the surface treatments, a timescale similar to that of the epoxy resin on Si substrates during the preparation of the interface specimens used for the interfacial analyses.

### Preparation of epoxy resin−Si substrate interfaces

DGEBA and PACM were mixed in a stoichiometric ratio of 1:1, corresponding to a molar ratio of 2:1 for DGEBA to PACM. These reagents were mixed using the following process: (i) a vial bin containing DGEBA and PACM was heated at 60 °C, above the melting temperatures of DGEBA (40–44 °C) and the glass transition temperature of PACM (30–60 °C), for 5 min; (ii) the melting mixture was stirred for 5 min using a homogenizer (Power Homogenizer Portable S-203, AS ONE

Corporation, Japan) which produces many bubbles; (iii) vacuum degassing was performed using a rotary pump at room temperature until the bubbles were removed. To prepare the adhesive interfaces, a mixture of DGEBA and PACM was applied to the Si substrates and subsequently cured at 100 °C for 90 min. The thicknesses of the applied epoxy resin were 190 μm, 20 μm, 20 μm, and 1 mm for the lap-shear test pieces, STEM-EELS specimens, TEM specimens for fracture-surface observations, and ATR-FTIR specimens, respectively. Because these thicknesses are sufficiently larger than the regions where the compositional changes and fractures are observed (within ~10 nm of the interfaces with the silicon substrates), the phenomena occurring near the interfaces should be common for all specimens.

The degree of cross-linking of the epoxy resins was determined to be ~85% using ATR-FTIR. The changes in the peak intensity at 910 cm$^{-1}$ (epoxy rings) during curing were standardized using the constant peak intensity at 830 cm$^{-1}$ (benzene rings), which remained unchanged throughout the curing process.

### FTIR measurement

Transmission FTIR and ATR-FTIR measurements were conducted using an FTIR Spectrometer Frontier (PerkinElmer, Inc., USA). The Si substrates used for ATR-FTIR analysis were obtained from R-DEC Co. To enable a fair comparison with the transmission FTIR spectra, the ATR-FTIR spectra were corrected by dividing them by the wavelength λ. This correction accounts for the fact that the absorbance in ATR-FTIR spectra is proportional to $d_p$ ($\propto \lambda$), as shown in equation S2 of the Supplementary Information. The OH- and H-terminated Si substrates were coated with the epoxy resin and cured using the aforementioned conditions.

### Lap shear tests

These tests were conducted to measure the adhesive strength between the epoxy resin and Si substrates. Two OH- and H-terminated Si substrates, each measuring 25 mm in width and 10 mm in length, were bonded to the DGEBA-PACM epoxy resin. A single-sided release film with a rectangular hole (25 mm width and 2 mm length) filled with a mixture of DGEBA and PACM was sandwiched between two Si substrates. The assembly was then heated at 100 °C for 90 min to cure the epoxy resin. The lapped region of the epoxy resin measured 25 mm in width, 2 mm in length, and 0.19 mm in thickness.

Furthermore, SUS304 plates measuring 25 mm in width, 100 mm in length, and 3 mm in thickness were bonded to both sides of the test pieces using a room-temperature curing adhesive. Small pieces consisting of two Si substrates, a release film, and a SUS plate were also attached to both ends of the SUS plates to adjust the thickness. The test pieces were mounted on a universal testing machine (Instron 5567, Instron Corporation, USA) with a 20 kN static load cell. The lap-shear tests were conducted at a tensile speed of 0.5 mm/min. The lap shear strength (adhesive strength) was calculated by dividing the load at the break by the lap area of 25 mm × 2 mm.

### STEM observation and STEM-EELS measurements

The ultrathin cross-sectional TEM specimens of the Si-epoxy interfaces were prepared by Ar ion milling at ~−170 °C using a Cryo Ion Slicer IB-09060CIS (JEOL Ltd., Japan). STEM images and EELS spectra, specifically C–K, N–K, and O–K edge ELNES, were acquired using a STEM system (JEM-ARM200F, JEOL Ltd., Japan). The microscope was equipped with a cold field emission source, aberration correctors for the probe- and image-forming lenses, and an EELS system (Enfinium ER, Gatan, Inc., USA) operated at 200 kV. The spectrometer was configured with a dispersion of 0.1 eV per channel, while the energy resolution of the EELS measurements was ~0.7 eV. Spectral imaging was conducted using a probe current of 30 pA, a probe size of ~0.1 nm, a dwell time of 4 s per pixel, a step size of 1 nm, and an image area of 5 pixels × 20 pixels (corresponding to 5 nm × 20 nm). A random sub-

pixel scan of 64 pixels was used to minimize electron irradiation damage. The irradiation damage had little impact on the major features of the C−K edge in the epoxy resin (DGEBA-PACM system) under the illumination condition. In particular, the peak intensities related to electron transitions from 1s to π* and σ* orbitals showed minimal changes[34]. The spectra obtained from the pixels were integrated along the direction parallel to the interface to improve the signal-to-noise ratio.

Si-L$_{2,3}$ edge ELNES was measured from the interfaces between the epoxy resin and the SiO$_2$ layer using a STEM (JEM-ARM200F, JEOL Ltd., Japan). The microscope was equipped with a Schottky emission source, a double Wien filter monochromator, aberration correctors for the probe-forming and image-forming lenses, and a Quantum ERS EELS system (Gatan, Inc., USA). The measurements were conducted at an acceleration voltage of 200 kV. The spectrometer was set at a dispersion of 0.1 eV/channel, and the energy resolution of the EELS measurements was ~0.3 eV. Spectral imaging was performed with a probe current of 3.7 pA, probe size of ~0.1 nm, dwell time of 1.7 s/pixel, step size of 1 nm/pixel, and image area of 5 pixel × 5 pixel (5 nm × 5 nm). To minimize electron irradiation damage, a random sub-pixel scan of 64 pixels was applied.

## Model preparation for ELNES simulations

To prepare stable structural models of SiO$_2$ surfaces with molecules, we performed structural optimization and molecular dynamics simulations based on density functional theory (DFT) calculations using Quantum Espresso software (ver. 6.4.1)[35]. We prepared OH- and H-terminated surface models (Fig. 3) based on a stable 96-atom SiO$_2$ dense surface, corresponding to the 2 × 2 unit-cells of an α-quartz (0001) dense surface model[36,37]. We applied the gauge-including-projector-augmented-wave method along with the Perdew–Burke–Ernzerhof exchange-correlation functional, specifically using pseudopotentials for {Si, O, and H}.pbe-tm-gipaw.UPF. To ensure accurate calculations, a k-point grid corresponding to the Brillouin zone gamma point was employed. For the plane-wave functions, a kinetic energy cut-off of 45.0 Ry was utilized, while a cut-off of 450.0 Ry was used for the charge density. To achieve stable structural models, we performed stabilization through a 500-fs run using DFT calculations at 300 K, consisting of 2000 MD steps with a time step of 0.25 fs.

To create grafted models of the H-terminated SiO$_2$ surface, we substituted the H atom with different molecules (-O-Ph, -O-C, -N, and -C), as depicted in Supplementary Fig. 6. Subsequently, structural optimizations and MD simulations were conducted using the same parameters and conditions as those applied to the OH-terminated and H-terminated surface models.

## ELNES simulations

The C-K edge ELNES spectra were simulated using the first principles plane wave basis pseudopotential method (CASTEP code)[38] on structure-optimized DGEBA and PACM molecules in 1.5 nm × 1.5 nm × 1.0 nm supercells. The structures used for the simulations are depicted in Fig. 4c. The C−K edges of all C sites within the models were individually calculated by incorporating the excited-state pseudopotential specific to each atomic site. The theoretical transition energies for all the spectra were determined using a method described in previous studies[39,40]. The individual spectra within each model were combined to generate the overall spectrum of the model. A Gaussian distribution with a full width at half maximum of 0.5 eV was applied to broaden the spectra.

Si-L$_{2,3}$ edge ELNES simulations were conducted on epoxy resin−silica interface models to analyse the chemical bonding states at the interfaces between the epoxy resin and Si substrate. The Si-L$_{2,3}$ edge corresponds to the electron transition from the Si-2p orbital, which is a shallower state compared to the C-1s orbital used for C−K edges. To perform these simulations, a different simulation code

called the first-principles all-electron orthogonalized linear combination of atomic orbital code was employed[41]. For the Si-L$_{2,3}$ edge simulations, different functional groups (-H, -OH, -O-Ph, -O-C, -N, and -C) were attached to a Si atom on the α-quartz (0001) dense surface (as shown in Fig. 3 and Supplementary Fig. 6). A vacuum layer with a thickness of ~1 nm was included in each model. The sizes of the models varied, ranging from ~1 × 1 × 2–3 nm, depending on the size of the adsorbed species. To accurately distinguish the spectral differences between the surface models, the Si-L$_{2,3}$ edge spectra of the Si atoms bonded to the additional species (-H, -OH, -O-Ph, -O-C, -N, and -C) were simulated by introducing a core hole into the Si-2p orbital. The theoretical transition energy was estimated by calculating the total energy difference between the ground and excited states.

## Curing molecular dynamics simulations

First, curing MD simulations for bulk systems without walls were performed to clarify the relationship between the stoichiometric ratios and cross-linked structures of the DGEBA−PACM epoxy resin system. To prepare the bulk systems with various stoichiometric ratios of 0.6, 0.8, 1.0, 1.2, and 1.4, the numbers of DGEBA and PACM molecules contained in the periodic cells were set to 420:126, 420:168, 420:210, 420:252, and 420:294, respectively. The DGEBA and PACM molecules were randomly packed in the cells with the procedure used in the previous studies[42,43].

Subsequently, curing MD simulations were conducted utilizing the reaction model proposed by Okabe et al.[24]. The reaction criteria for the curing MD simulations were based on the distance between the reaction sites and the Arrhenius-type reaction probabilities calculated using activation energies and local temperatures. After the reaction, the heat of formation was released as kinetic energy at the reaction sites. Further details on the procedure for curing MD simulations can be found in previous reports[24,42,43]. In this study, a distance criterion of 5.64 Å was used. The activation energy and heat of formation for the primary amine-epoxy reaction were 42.72 and 21.92 kcal mol$^{-1}$, respectively. For the secondary amine-epoxy reaction, the activation energy and heat of formation were 31.94 and 19.45 kcal mol$^{-1}$, respectively. The activation energies and heat of formation were computed using global reaction route mapping (GRRM)[44–47] at the B3LYP/6-31 G(d) theoretical level. Annealing and cooling simulations were performed to prepare a cured resin model at room temperature[25,48]. The DREIDING force fields were used for the polymers[49]. Partial atomic charges were assigned using QEq[50–52]. The parameters suggested by Lopes et al. were employed along with the CHARMM force field for the silica walls[53]. All MD simulations were performed using LAMMPS[54].

Finally, curing MD simulations were conducted to investigate the behaviour of epoxy resins confined between silica walls with different surface chemistries. First, as depicted in Fig. 7a, the (10$\bar{1}$0) surfaces of α-quartz SiO$_2$ were terminated with OH and H groups, representing the SiO$_2$ layers. These surfaces exhibit periodic structures in the x- and y-directions and a surface structure in the z-direction. Subsequently, a random arrangement of DGEBA and PACM molecules was introduced between the two silica surfaces. It is difficult to completely reproduce the species distribution inside the adhesive layer in a small system such as MD simulations. To reproduce the stoichiometric ratios near the interface obtained from EELS, curing MD simulations were performed by changing the overall stoichiometric ratio for the OH-terminated and H-terminated systems, respectively. For the OH-terminated system, the overall stoichiometric ratio was set to 1.4 using 720 DGEBA and 504 PACM molecules. For the H-terminated system, 720 DGEBA and 288 PACM molecules were used, and the stoichiometric ratio was set to 0.8. Initially, the cell dimensions were set to 58.9 Å × 63.6 Å × 851.5 Å, and the temperature was set to 327 °C. The MD cell was then compressed along the z-direction until the cell length reached 251.5 Å, while maintaining a constant temperature. Following compression, the system was cooled to the curing temperature of 100 °C under constant

volume conditions. Finally, an NPT simulation was performed for 2 ns at 100 °C and a pressure of 1 atm to allow for relaxation. The following curing simulations were conducted with the same conditions as those used for the bulk systems described above.

### TEM observation of fracture surfaces

Thin cross-sectional sections of the OH- and H-terminated interfaces were obtained using a FIB system (JEM-9310FIB, JEOL Ltd., Japan). Each section had a thickness of ~300 nm. The sections were then mounted on tensile cartridges, positioned to bridge the slit (width of ~5 μm), ensuring that the interfaces were located inside the slit. The interfacial specimens were subjected to tensile loading and fracture at a rate of ~100 nm s$^{-1}$ using the tensile TEM holder. Following the fracturing process, the TEM holder was inserted into a TEM JEM-F200 (JEOL Ltd., Japan). Cross-sections of the fracture surfaces were observed using TEM.

### Data availability

Source data of the ADF-STEM images are provided with this paper. The atomic coordinates of the cured bulk structures of epoxy resin with various stoichiometric ratios generated by the curing MD simulations are provided as Supplementary Data 1. The atomic coordinates of the OH- and H-terminated interfaces generated by the curing MD simulations are provided as Supplementary Data 2. Source data are provided with this paper.

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

## Acknowledgements

This work was supported by the Japan Science and Technology Agency (JST) CREST to H.J. (Grant Numbers JPMJCR1993 and JPMJCR19T4) and the Japan Society for the Promotion of Science (JSPS) KAKENHI (Grant Numbers 20K15330 and 23H02017 to T.M. and 22H00329 to H.J.). STEM observations and EELS measurements were performed at the Tohoku University Microstructural Characterization Platform and Kyoto University Microstructural Characterization Platform in the Nanotechnology Platform Project, sponsored by the Ministry of Education, Culture, Sports, Science, and Technology (MEXT), Japan. We would like to thank Editage (www.editage.com) for English language editing.

## Author contributions

The manuscript was written through contributions of all authors. All authors approved the final version of the manuscript. T.Miy., Y.K.S., and H.J. wrote the manuscript; T.Miy. and H.J. coordinated the experiments; T.Miy., Y.K.S., S.K., and M.M. prepared the specimens; T.Miy., S.K., and K.S. performed the lap-shear test; T.Miy. and Y.K. performed the STEM-EELS analysis; H-F.W. and A.K. performed the TEM observation; Y.K. and T.O. did the curing MD simulation; K.H. and T.Miz. contributed the ELNES simulation; T.Miy., Y.K.S., and M.M. contributed the ellipsometry, contact angle measurements, and ATR-FTIR measurements; K.S. contributed the finite element analysis; K.Y. and H.-H.H. helped interpret the results; H.J. supervised the project.

## Competing interests

The authors declare no competing interests.

## Additional information

[1]Institute of Multidisciplinary Research for Advanced Materials, Tohoku University, Sendai, Miyagi 980-8577, Japan. [2]Department of Aerospace Engineering, Graduate School of Engineering, Tohoku University, 6-6-01 Aramaki Aza Aoba, Aoba-ku, Sendai, Miyagi 980-8579, Japan. [3]Department of Finemechanics, Graduate School of Engineering, Tohoku University, 6-6-01 Aramaki Aza Aoba, Aoba-ku, Sendai, Miyagi 980-8579, Japan. [4]Department of Chemical and Materials Engineering, National Central University, No. 300, Zhongda Rd., Zhongli Dist., Taoyuan City 320317, Taiwan. [5]Department of Applied Chemistry, Graduate School of Engineering, Tohoku University, 6-6-07 Aramaki Aza Aoba, Aoba-ku, Sendai, Miyagi 980-8579, Japan. [6]New Industry Creation Hatchery Center, Tohoku University, Sendai, Miyagi 980-0845, Japan. [7]Nanostructures Research Laboratory, Japan Fine Ceramics Center, Nagoya, Aichi 456-8587, Japan. [8]Research Center for Structural Materials, Polymer Matrix Hybrid Composite Materials Group, National Institute for Materials Science, 1-2-1 Sengen, Tsukuba, Ibaraki 305-0047, Japan. [9]Department of Materials Science and Engineering, University of Washington, BOX 352120, Seattle, WA 98195, USA. [10]Department of Applied Physics, National Defense Academy, Yokosuka, Kanagawa 239-0811, Japan. [11]Institute of Industrial Science, The University of Tokyo, Meguro-ku, Tokyo 153-8505, Japan. ✉e-mail: kawagoe@tohoku.ac.jp; keiichi.shirasu.c1@tohoku.ac.jp; hiroshi.jinnai.d4@tohoku.ac.jp

