## [Peer Review File · Nature Communications]

Effect of Inorganic Material Surface Chemistry on Structures and Fracture Behaviours of Epoxy ResinREVIEWER COMMENTS

Reviewer #1 (Remarks to the Author):

This manuscript lacks innovation and cannot generate widespread academic interest, far from reaching the level of papers that NC can accept.

Reviewer #2 (Remarks to the Author):

This is a fascinating manuscript, putting forth interesting new arguments about the role of surface chemistry in adhesion and fracture of epoxy amine resins, particularly the role of condensed amine at the interface.

The work is thorough, and an unusually broad range of techniques have been deployed to investigate the nature of the interphase / fracture location in these materials. There are however, some issues which should be addressed prior to publication:

- 1, First, the authors should include more detail about the application of the epoxy-amine - by what method was this performed? What are the film thicknesses achieved? How long were the precursors mixed prior to application? Each of these can independently impact the film composition.
- 2, Second, the discussion around entropic segregation is somewhat confusing. The authors state that this is expected due to the size difference between DGEBA and PACM, yet this effect seems to be absent on hydrogen terminated silicon.
- 3, Importantly, the lap shear results do not seem to be statistically significant.
- 4, Some discussion / justification of the reproducibility of ATR-FTIR results is needed, from experience, fluctuations in the background can lead to such minor changes in band intensity. In addition, the plot presented in Figure S4 does not follow the convention of higher to lower frequency on the x-axis.
- 5, A scheme illustrating the proposed structure of the interface might be helpful - rather confusingly, on the one hand the authors state that PACM condenses at the Si-OH interface, on the other hand, MD simulations seem to show that tertiary amines are more prevalent (indicating that the network is more cross-linked in this region).
- 6, Lastly, the abstract, in particular the first and final sentences, should be reworded as it is unclear. The motivation and main conclusions are not successfully summarised.

Reviewer #3 (Remarks to the Author):

This is a very interesting and well thought out paper. The authors seek to better understand the mechanisms of adhesion between epoxy resin and an inorganic filler, motivated by the desire to replace conventional metallic structures with light weight composite materials, or mechanical fasteners such as bolts or rivets with epoxy adhesives. In these applications, the integrity of the interface between the organic epoxy and the inorganic filler or substrate is important for maximizing the mechanical performance of the system. Epoxies are typically a two (or more) component mixture of a monomeric epoxy that contains glycidyl rings that are designed to react with a multi-functional cross-linking agent. The functionality of the cross-linking agent is typically primary amines. These components are mixed and under the action of heat and catalyst, the system forms a highly cross-linked network. How this cross-linked network interacts with or is affected by the inorganic filler has been a long issue in the composite's community. It is generally appreciated that the nature of the interactions at the filler-matrix interface plays a significant role in the mechanical performance of the composite. But the chemical nature of this interface has been difficult to quantify.

The authors shed light on this topic through analytical TEM and STEM methods to characterize the chemistry and composition at the interface. They create model systems based on a very common and well-studied epoxy system, DGEBA (epoxy) cross-linking with PACM (diamine). For inorganic filler, they glue together very smooth Si wafers with the epoxy resin as a model interface. Using standard semiconductor processing schemes, they create two versions of the Si interface: an -OH rich hydrophilic interface and an -H terminated analog that is hydrophobic. Otherwise, the mechanical roughness the two interfaces are identical. This is a nice extension of a previously published study where the authors created similar model interfaces using aluminum substrates that convoluted variations in the mechanical roughness of the interface with changing in the surface chemistry (ref 18); this previous reference also used analytical TEM and STEM to quantify the interfacial chemistry and composition but could not deconvolute the effects of roughness from their findings. To study these model interfaces, the authors perform three primary types of experiments. First, they perform macroscopic lap-shear tensile tests to quantify the bonding strength of the epoxy to the -OH and -H terminated surfaces. Second, they perform STEM-EELS measurements and modeling across the model interfaces between the Si substrate and epoxy adhesive to quantify the composition of the DGEBA and PACM components as a function of distance away from the substrate. Lastly, they attempt to corroborate the results of the experimental composition maps using a reactive atomistic MD simulation package that has been developed for cross-linked epoxy. The approach is well-thought out, methodical, and well executed. I like the study.

The authors key findings are that adhesive bond strength of the cross-linked epoxy resin is slightly stronger to the -OH terminated Si substrate than the -H terminated analog. The slight increase in adhesion correlates a higher ratio in the concentration of the PACM/DGEBA components in the epoxy network in the few nanometers near the -OH interface. This preferential enrichment of the PACM component near the -OH interfaces was supported by EELS imaging at both the N-K edge (only present in PACM), the π^* peak at the C-K edge (only present in the DGEBA), macroscopic wetting studies of the individual components (PACM, DGEBA, and an unreacted mixture of PACM+DGEBA) on the two substrates, and the reactive MD modeling studies. Taken in total, the authors have convinced me that there is a slight enrichment in the PACM component of the epoxy mixture near the -OH terminated Si

substrate that seems to correlate with a slight increase in adhesion. This is an important finding. However, I also have some minor issues that prevent me from recommending publication as the manuscript currently stands. I will articulate these issues below. In my view these need to be adequately addressed before proceeding.

1. The observed effects are subtle, and potentially not supported by the uncertainty in the data. All of the trends seem to go in a consistent direction, but the uncertainty is large. The enhancement of the adhesive strength of the epoxy to the -OH surface is 1.7 MPa (30.9 vs 32.6 MPa), but the enhancement is smaller than the uncertainty of the measurements (2.1 MPa); technically, the two measurements are within error. Is there a way to decrease the error bars on these measurements? This is a little bit troublesome for a high impact journal like Nature Materials. The enhancement in the N-K/C-K composition near the -OH interface (Figure 5a) seems to be larger than the uncertainty, but only in the first data point 1 nm away from the interface. At 2 nm and beyond, the compositions are within error. The entire notion of enhanced composition seems to be based on this single datapoint. In figure 4c and 4d, the authors show raw spectra of the C-K edge and claim there is qualitative decrease in the π^* peak near the -OH surface that would support enrichment of the PACM, but those peak (or shoulder) heights are not quantified. Why did they not quantify this enhancement? I would really like to see a stronger quantitative basis for this enrichment.

2. While I tend to believe there is some form of enrichment of the PACM near the -OH interfaces, I am not entirely sure the chemical mechanisms are that lead to the enhanced mechanical properties. Generally, primary amines (-NH₂) in the PACM should not react with the silanol (Si-OH) groups at the interface. The two groups both too nucleophilic. So, what is the mechanism by which the cross-linking density increases near the interface? Just the enrichment of PACM near the surface alone would not seemingly lead to increased cross-link density. Is this a catalytic effect? It is well known that -OH groups can catalyze the reaction between -NH₂s and epoxides. Is it more the presence of the surface -OH's that drives the reaction to a greater extent near the surface than the excess PACM that is responsible for their observations? Enrichment of PACM near the interface alone would seem to drive the composition off-stoichiometry and actually reduced the cross-link density.

3. The mechanical tests need to be explained in a little greater detail. The macroscopic lap shear experiments clearly indicate the interface was loaded in shear. However, the in-situ TEM experiments seem to be loading that interface in tension? The manuscript is not clear about this detail. Were the two loading conditions the same? Or different? If different, how does this play into the interpretation?

4. The TEM fracture data presented in Figure 8 is used to argue that there is more cohesive failure in the -OH samples while the -H samples exhibit more interfacial delamination. Clarifications to point 3 above will help interpret these results. I would agree that there does seem to be more mechanical drawing of the resin near the -OH substrate, and more clean fracture near the -H substrate, but I need to know more details about the test. In bulk DGEBA / PACM resin, in the process zone of a crack near the initiation site, where the crack is slowly growing and damage is building up, you can see local drawing of

the material. However, once the crack reaches its critical length and starts to propagate catastrophically, the fast fracture surfaces look smooth. This could also be an explanation for the difference between Figures 8c and 8d. I need to know more about where these images came from with regards to the process zone and fast fracture zone to have confidence in the authors assertion.

We thank all reviewers for their efforts in reviewing our manuscript, and for providing valuable insights that helped us improve the quality of our paper. Please note that modifications are highlighted in **red** in the revised manuscript and supporting information.

Reviewer #1

Comment 1:

This manuscript lacks innovation and cannot generate widespread academic interest, far from reaching the level of papers that NC can accept.

Reply 1:

The following discussion explains the innovative points and widespread academic interest in our study. Although adhesion and delamination are essential from a fundamental and industrial point of view, they are still immature research fields. Many existing studies are merely phenomenological. For example, it is widely known that adhesion depends on the chemical interaction between two materials and the surface morphological effect, the so-called “anchoring effect.” However, most previous studies did not separate these two effects or clarify their relationship. We aimed to investigate purely chemical interactions without the anchoring effect. To do this, an understanding of the relationship between the adhesion strength and interfacial structure *at the molecular level* is required. Our study is the first to clarify the structure with sub-nm-resolution around the organic–inorganic interfaces. The origin of adhesion strength was successfully identified based on the detailed molecular-level structural information. We believe that our study is pioneering fundamental research in adhesion and delamination at the molecular level, which could eventually be applied to industrial products.

In addition, the combination of advanced electron microscopy and molecular simulations is an advantage over similar previous studies. Preparing ultra-thin films of several tens of nanometers from hard/soft composites and performing electron energy loss spectroscopy for quantitatively determining stoichiometric amine-to-epoxy ratios with nanometer resolution are challenging. The molecular simulations were performed in a data-assimilation manner to further “visualize” the cross-linked structure of epoxy resin near the hard/soft interface. We note that the cross-linked structure cannot be visualized by microscopy and can only be visualized by data assimilation. We believe that these aspects are significantly novel and impactful to warrant publication in Nature Communications.

Reviewer #2

This is a fascinating manuscript, putting forth interesting new arguments about the role of surface chemistry in adhesion and fracture of epoxy amine resins, particularly the role of condensed amine at the interface. The work is thorough, and an unusually broad range of techniques have been deployed to investigate the nature of the interphase / fracture location in these materials. There are however, some issues which should be addressed prior to publication:

Reply:

We greatly appreciate the reviewer's careful reading of our manuscript and valuable comments, which have been very helpful in improving our manuscript.

Comment 1:

1, First, the authors should include more detail about the application of the epoxy-amine - by what method was this performed? What are the film thicknesses achieved? How long were the precursors mixed prior to application? Each of these can independently impact the film composition.

Reply 1:

We agree that the detailed mixing and curing processes are essential information for discussing the properties and cross-linked structures of epoxy resin, so further details were added in the Methods section.

Page 29, lines 483-497:

The epoxy resin prepolymer and curing agent were mixed in a stoichiometric ratio of 1:1, corresponding to a molar ratio of 2:1 for DGEBA to PACM. **These reagents were mixed using the following process: (i) a vial bin containing DGEBA and PACM was heated at 60 °C, above the melting temperatures of DGEBA (40–44 °C) and the glass transition temperature of PACM (30–60 °C), for 5 min; (ii) the melting mixture was stirred for 5 min using a homogenizer (Power Homogenizer Portable S-203, AS ONE Corporation, Japan) which produces many bubbles; (iii) vacuum degassing was performed using a rotary pump at room temperature until the bubbles were removed.** To prepare the adhesive interfaces, a mixture of DGEBA and PACM was applied to the Si substrates and subsequently cured at 100 °C for 90 min. **The thicknesses of the applied epoxy resin were 190 μm, 20 μm, 20 μm, and 1 mm for the lap-shear test pieces, STEM-EELS specimens, TEM specimens for fracture-surface observations, and ATR-FTIR specimens, respectively. Because these thicknesses are sufficiently larger than the regions where the compositional changes and fractures are observed (within ~10 nm of the interfaces with the silicon substrates), the phenomena occurring near the interfaces should be common for all specimens.**

Comment 2:

2, Second, the discussion around entropic segregation is somewhat confusing. The authors state that this is

expected due to the size difference between DGEBA and PACM, yet this effect seems to be absent on hydrogen terminated silicon.

Reply 2:

The manuscript has been revised to clarify the discussion of these points. In fact, we were not able to reveal the interfacial segregation mechanisms of amine molecules in this study. The only possible way to visualize the diffusion of molecules during curing is by molecular dynamics simulations. However, reproducing the complex experimental conditions in such simulations is currently impossible, mainly due to the small system sizes and short simulation times.

In the original manuscript, entropic segregation was described as a conceivable mechanism for the interfacial enrichment of PACM. However, as the reviewer pointed out, this entropic effect in our interfacial systems is not certain. Thus, we removed the description for clarity. As described later, the experimental results themselves, *i.e.*, the fact that the stoichiometric ratios of PACM and DGEBA vary near interfaces, remain unchanged.

Page 14, lines 238-242:

~~A previous study reported the entropic condensation of small molecules near solid interfaces.²⁴ In the case of the DGEBA-PACM system, where PACM molecules are smaller than DGEBA molecules, PACM molecules that are not engaged in the cross-linked network have the potential to condense at the interface through diffusion during the curing process, driven by the entropy effect.~~

Comment 3:

3, Importantly, the lap shear results do not seem to be statistically significant.

Reply 3:

We agree that the statistical significance of the lap shear test is critical to our study. In fact, due to the brittleness of the silicon substrates, the success rate of the lap shear test is very low, and thus, the statistical accuracy was poor in the original manuscript. According to the reviewer's comment, we performed the lap shear tests again with more test pieces (30 for each OH- and H-terminated interface). As a result, we obtained statistically significant results for OH-terminated interface: 34.8 ± 1.2 MPa (N=8) and H-terminated interface: 30.8 ± 1.8 MPa (N=5). These values have been updated in the main text. In addition, as a reference, we added the pictures of successful and failed lap shear pieces as follows.

Pages 6-7, lines 111-113:

Lap-shear tests were conducted to measure the adhesive strengths of the OH- and H-terminated interfaces, yielding values of 34.8 ± 1.2 ~~32.6 ± 2.1~~ and 30.8 ± 1.8 ~~30.9 ± 2.1~~ MPa, respectively.

Supporting Information, page 16 Figure S12:

Figure S12. Schematics of the lap-shear test and successful and failed pieces.

Comment 4:

4. Some discussion / justification of the reproducibility of ATR-FTIR results is needed, from experience, fluctuations in the background can lead to such minor changes in band intensity. In addition, the plot presented in Figure S4 does not follow the convention of higher to lower frequency on the x-axis.

Reply 4:

To justify the differences in peak intensities observed in the ATR-FTIR spectra of the OH- and H-terminated interfaces, we show a wide-range spectrum ($2400\text{--}3600\text{ cm}^{-1}$) in Figure S4. This figure indicates that the background fluctuations are small (e.g., see the ranges of $2400\text{--}2600\text{ cm}^{-1}$ and $3120\text{--}3160\text{ cm}^{-1}$). In addition, the intensity difference between the $\nu_s(\text{CH}_2)$ and $\nu_{as}(\text{CH}_2)$ peaks in the spectra of the OH- and H-terminated interfaces is more than ten times larger than that in the background regions. Therefore, we conclude that the intensity differences of the vibrational peaks are significant. The transmission IR spectra in the original manuscript were removed due to the extensive background fluctuations. An ATR-FTIR spectrum obtained from the interface between the epoxy resin and piranha solution-treated silicon substrate was added for comparison to confirm the reproducibility. This interface also has an OH-termination, and its spectrum almost perfectly overlaps with that of the OH-terminated interface over a wide range, which ensures the reproducibility of the spectrum of the OH-terminated interface. Finally, we reversed the wavenumber-axis following the reviewer's advice. The text in the Supporting Information was also revised.

Supporting Information, page 6, lines 120-121:

The dotted black lines in Figure S4 represent the ATR-FTIR spectrum of an interface between the epoxy resin and a silicon substrate treated with piranha solution, which also has an OH termination.

Supporting Information, pages 6-7, lines 128-135:

The OH-terminated interface spectrum exhibits higher intensities of the CH₂ vibrational peaks compared to those in the H-terminated interface spectrum. The intensity fluctuations and the discrepancies between the spectra of the OH- and H-terminated interfaces in the background regions (e.g., 2400–2600 cm⁻¹ and 3120–3160 cm⁻¹) are more than ten times smaller than the intensity difference of the $\nu_s(\text{CH}_2)$ and $\nu_{as}(\text{CH}_2)$ peaks. Moreover, the spectrum of the interface between the epoxy resin and an OH-terminated Si substrate (treated with piranha solution) exhibited an excellent agreement with that of the OH-terminated interface. These findings strongly highlight the significance of the observed difference between the OH and H-terminated interfaces.

Supporting Information, page 8:

Figure S4. ATR-FTIR spectra of the OH- and H-terminated interfaces, along with that of an interface between the epoxy resin and a Si substrate treated with piranha solution, which also has an OH-terminated interface. The absorbance of the ATR-FTIR spectra was adjusted to enable comparison with the transmission FTIR spectrum. The absorbances of all spectra were normalized based on the intensity of the peaks corresponding to the benzene rings and epoxy groups (3020–3080 cm⁻¹) present in DGEBA.

Comment 5:

5, A scheme illustrating the proposed structure of the interface might be helpful - rather confusingly, on the one hand the authors state that PACM condenses at the Si-OH interface, on the other hand, MD simulations seem to show that tertiary amines are more prevalent (indicating that the network is more cross-linked in this region).

Reply 5:

In the original manuscript, the correspondence between the experimental and simulation results was premature because the STEM-EELS analysis and MD simulation were performed *independently*. In this revision, we performed the curing MD simulations *based on the experimentally measured stoichiometric ratios (amine-to-epoxy ratios) near the interfaces* to explore the cross-linked structures near the interfaces

with the specific chemical compositions of the epoxy resin. In this way, the simulation results are more relevant for comparison with the experimental results (based on the stoichiometric ratios).

We performed simulations for two types of systems: (i) bulk systems with different stoichiometric ratios and (ii) interfacial systems reflecting the near-interface stoichiometric ratios estimated by the STEM-EELS analysis. These simulations revealed the following findings: (i) Larger linear secondary amines and hydroxyl groups are formed within the epoxy resin near the OH-terminated interface owing to the larger stoichiometric ratio (amine concentration) than the H-terminated interface. (ii) The number densities of the branched structures (tertiary amines) exhibited a small difference between the OH- and H-terminated interfaces. (iii) Hydrogen bonds are formed between the hydroxyl groups generated in the epoxy resin and the silanol groups on the OH-terminated surface, resulting in a higher interfacial adhesion energy. Furthermore, we added the characteristics of the cross-linked and adsorption structures near the OH- and H-terminated interfaces in Figure 9. The manuscript was revised as follows.

Pages 14-15, lines 243-254 and Figure 5:

The profiles of the stoichiometric ratio (amine-to-epoxy ratio) were evaluated by dividing the π^* peak intensity by the σ^* peak intensity in the C-K edge spectrum, as described in Section S10 in the Supporting Information. These profiles in Figure 5 (red circles) show that the stoichiometric ratio at the OH-terminated interface (~ 1.3) is larger than that in the bulk (1.0). In contrast, the epoxy resin close to the H-terminated interface shows a stoichiometric ratio of ~ 0.7 , which is smaller than that of the bulk region.

Figure 5. Profiles of the stoichiometric ratio (amine-to-epoxy ratio) measured at different distances from the (a) OH-terminated and (b) H-terminated adhesive interfaces. The red circles represent the stoichiometric ratio estimated from the intensity of the π^* peak divided by that of the corresponding σ^* peak of the C-K

edge spectrum (C-K π^*/σ^*). The blue squares indicate the integrated intensity of the N-K edge spectrum divided by that of the corresponding C-K edge spectrum (N-K/C-K).

Page 15, lines 256-257:

To verify the stoichiometry changes near the interfaces, compositional analysis of the epoxy resin was conducted using N-K-edge ELNES of the OH- and H-terminated interfaces.

Page 15, lines 266-272:

The blue squares in Figures 5(a) and 5(b) depict the stoichiometric ratios calculated from the N-K/C-K intensities plotted against the distance from the OH- and H-terminated interfaces, respectively. In the case of the OH-terminated interface, the stoichiometric ratio of PACM is higher near the oxide-layer interface compared to the bulk region of the epoxy resin. Conversely, a gradual decrease in the stoichiometric ratio is observed for the H-terminated interface from the bulk toward interfacial regions. These profiles are consistent with those obtained from the C-K edge analysis.

Pages 16, lines 281-291 and Figure 6:

Curing molecular dynamics simulation of bulk systems of epoxy resins

The variations in the stoichiometric ratios of DGEBA to PACM should result in different cross-linked structures. To evaluate the cross-linked structures of the epoxy resin near the interfaces, we conducted curing MD simulations that consider the formation of cross-linked structures through exothermal reactions.²⁴ Figure 6(a) shows an example of a periodic cell model used to investigate the cross-linked structures formed in bulk systems with different stoichiometric ratios. The simulation considered the primary and secondary amine-epoxy reactions (Figure S5), forming linear secondary and branched tertiary amines and hydroxyl groups (Figure 6(b)). Figure 6(c) shows that the number density of the tertiary amines (branched structures) is almost independent of the stoichiometric ratio, while those of the secondary amines (linear structures) and hydroxyl groups increase with increasing stoichiometric ratio.

Figure 6. Curing molecular dynamic simulations of bulk systems of DGEBA-PACM epoxy resins with various stoichiometric ratios. (a) Model of the bulk system used for the simulations. (b) Schematics of the secondary and tertiary amine structures. Hydroxyl groups are generated by the reactions to form secondary and tertiary amines. (c) Number densities of the secondary and tertiary amines and hydroxyl groups as a function of the stoichiometric ratio.

Pages 17-20, lines 299-336 and Figure 7:

Curing molecular dynamics simulation on epoxy resin-silica interfaces

Subsequently, curing MD simulations of interfacial models were performed to explore the cross-linked and adhesion structures close to the OH- and H-terminated interfaces in detail. In the simulation, the (10 $\bar{1}$ 0) surfaces of α -quartz were terminated with OH and H as models of the oxidized Si layers, as shown in Figure

7(a). The uncured epoxy resin, consisting of DGEBA and PACM molecules, was confined between the OH- and H-terminated surfaces. The average stoichiometric ratios of the uncured epoxy resins within the OH- and H-terminated systems were set to reflect the interfacial compositions measured by STEM-EELS. Then, curing MD simulations were performed on these systems. Note that the simulations did not consider chemical reactions between the epoxy resin molecules and the substrates.

The stoichiometric ratios of the cured epoxy resins within 1 nm of the OH- and H-terminated interfaces (grey region in Figure S11) are approximately 1.5 and 1.0, respectively, which are similar to the experimentally measured ratios. Figure 7(b) depicts the number density distributions of the reacted amino groups (secondary and tertiary amines) in the epoxy resins near the OH- and H-terminated interfaces. Within 1 nm of the interface (grey regions), the number densities of tertiary amines (branched structures) are almost equal for both the OH- and H-terminated interfaces, whereas the OH-terminated interface, with a higher stoichiometric ratio, has more secondary amines (linear structures) than the H-terminated interface. This tendency is in good agreement with that of bulk systems, as shown in Figure 6(c). In other words, the OH-terminated interface has a higher average molecular weight between the cross-links than the H-terminated interface. The presence of distinct cross-linked structures at the interfaces could result in different mechanical properties.²⁵

The red lines in Figure 7(b) show the distributions of the hydroxyl groups, originating from the reactions between epoxy and amino groups, as shown in Figure S5. The OH-terminated system exhibits a higher average density of the hydroxyl groups in the epoxy resin compared to the H-terminated system. This originates from the higher stoichiometric ratio in the OH-terminated system, as observed for the bulk system (Figure 6(c)). Furthermore, the adsorption layers of hydroxyl groups exhibit a higher density near the OH-terminated interface compared to that near the H-terminated interface (denoted by black triangles). This difference arises from the formation of hydrogen bonds between the hydroxyl groups on the silica surface (silanol groups) and the hydroxyl groups generated within the epoxy resin.

Figure 7(c) illustrates the simulated atomic structure of the OH- and H-terminated interfaces. The hydrogen bonds (hydrogen-oxygen distance: <0.25 nm) between the silanol groups of the substrate and hydroxyl groups of the epoxy resin are indicated by green lines. The illustration indicates that numerous hydrogen bonds are formed at the OH-terminated interface, while there are very few hydrogen bonds at the H-terminated interface. Consequently, the OH-terminated interface with a PACM-rich composition generates many hydroxyl groups in the epoxy resin and preferentially forms hydrogen bonds to stabilize the adsorption structure.

Figure 7. Epoxy resin structures cured at the OH- and H-terminated interfaces. (a) Simulation model before curing and the surface structures of the OH- and H-terminated SiO₂ (α -quartz) used for the curing molecular dynamics simulations. The top faces of the SiO₂ crystals are terminated with hydroxyl groups and hydrogen atoms, respectively. (b) Number densities of the secondary and tertiary amines and hydroxyl groups in the epoxy resins near the OH- and H-terminated interfaces. The number densities of Si and O atoms in the substrates are shown by black dotted lines to clarify the positions of the interfaces. (c) Schematics of the OH- and H-terminated interfaces.

Page 21, line 356-357:

The E_{ad} values estimated for the OH- and H-terminated interfaces from the curing MD simulations were 0.223 and 0.147 J/m² ~~0.218 and 0.153 J/m²~~, respectively.

Page 25, Figure 9:

Figure 9. Schematics of the interfacial structures and fracture paths near the (a) OH-terminated and (b) H-terminated interfaces.

Comment 6:

6, Lastly, the abstract, in particular the first and final sentences, should be reworded as it is unclear. The motivation and main conclusions are not successfully summarised.

Reply 6:

The reviewer's comment made us reconsider our study's novelty. The abstract was revised accordingly to highlight the novelty and main conclusions.

Page 3, Abstract:

~~Adhesion and delamination between polymers and inorganic materials are being widely utilized. However,~~ The mechanisms underlying the influence of inorganic material surface chemistry on polymer structures and fracture behaviours near adhesive interfaces are not fully understood; this was studied herein using electron microscopy and molecular dynamics simulations. We prepared adhesive interfaces between epoxy resin and silicon substrates with varying surface chemistries (OH and H terminations) with a smoothness of <1 nm. The epoxy resins within sub-nanometre distance from the adhesive interfaces exhibited distinct amine-to-epoxy ratios, cross-linked network structures, and adhesion energies. The OH- and H-terminated interfaces exhibited cohesive failure and interfacial delamination, respectively. The substrate surface chemistry impacted the cross-linked structures of the epoxy resins within several nanometres of the interfaces ~~and the adsorption structures of molecules at the interfaces~~, which resulted in different fracture

behaviours and adhesive strengths. ~~This will help develop durable adhesion with epoxy resins, and potentially help achieve weight reduction in vehicles and other products.~~

Reviewer #3

This is a very interesting and well thought out paper. The authors seek to better understand the mechanisms of adhesion between epoxy resin and an inorganic filler, motivated by the desire to replace conventional metallic structures with light weight composite materials, or mechanical fasteners such as bolts or rivets with epoxy adhesives. In these applications, the integrity of the interface between the organic epoxy and the inorganic filler or substrate is important for maximizing the mechanical performance of the system. Epoxies are typically a two (or more) component mixture of a monomeric epoxy that contains glycidyl rings that are designed to react with a multi-functional cross-linking agent. The functionality of the cross-linking agent is typically primary amines. These components are mixed and under the action of heat and catalyst, the system forms a highly cross-linked network. How this cross-linked network interacts with or is affected by the inorganic filler has been a long issue in the composite's community. It is generally appreciated that the nature of the interactions at the filler-matrix interface plays a significant role in the mechanical performance of the composite. But the chemical nature of this interface has been difficult to quantify.

The authors shed light on this topic through analytical TEM and STEM methods to characterize the chemistry and composition at the interface. They create model systems based on a very common and well-studied epoxy system, DGEBA (epoxy) cross-linking with PACM (diamine). For inorganic filler, they glue together very smooth Si wafers with the epoxy resin as a model interface. Using standard semiconductor processing schemes, they create two versions of the Si interface: an -OH rich hydrophilic interface and an -H terminated analog that is hydrophobic. Otherwise, the mechanical roughness the two interfaces are identical. This is a nice extension of a previously published study where the authors created similar model interfaces using aluminum substrates that convoluted variations in the mechanical roughness of the interface with changing in the surface chemistry (ref 18); this previous reference also used analytical TEM and STEM to quantify the interfacial chemistry and composition but could not deconvolute the effects of roughness from their findings. To study these model interfaces, the authors perform three primary types of experiments. First, they perform macroscopic lap-shear tensile tests to quantify the bonding strength of the epoxy to the -OH and -H terminated surfaces. Second, they perform STEM-EELS measurements and modeling across the model interfaces between the Si substrate and epoxy adhesive to quantify the composition of the DGEBA and PACM components as a function of distance away from the substrate. Lastly, they attempt to corroborate the results of the experimental composition maps using a reactive atomistic MD simulation package that has been developed for cross-linked epoxy. The approach is well-thought out, methodical, and well executed. I like the study.

The authors key findings are that adhesive bond strength of the cross-linked epoxy resin is slightly stronger

to the -OH terminated Si substrate than the -H terminated analog. The slight increase in adhesion correlates a higher ratio in the concentration of the PACM/DGEBA components in the epoxy network in the few nanometers near the -OH interface. This preferential enrichment of the PACM component near the -OH interfaces was supported by EELS imaging at both the N-K edge (only present in PACM), the π^* peak at the C-K edge (only present in the DGEBA), macroscopic wetting studies of the individual components (PACM, DGEBA, and an unreacted mixture of PACM+DGEBA) on the two substrates, and the reactive MD modeling studies. Taken in total, the authors have convinced me that there is a slight enrichment in the PACM component of the epoxy mixture near the -OH terminated Si substrate that seems to correlate with a slight increase in adhesion. This is an important finding. However, I also have some minor issues that prevent me from recommending publication as the manuscript currently stands. I will articulate these issues below. In my view these need to be adequately addressed before proceeding.

Reply:

We appreciate the reviewer's careful reading of our original manuscript and the valuable comments, which are very helpful in strengthening our manuscript.

Comment 1-1:

1. The observed effects are subtle, and potentially not supported by the uncertainty in the data. All of the trends seem to go in a consistent direction, but the uncertainty is large. The enhancement of the adhesive strength of the epoxy to the -OH surface is 1.7 MPa (30.9 vs 32.6 MPa), but the enhancement is smaller than the uncertainty of the measurements (2.1 MPa); technically, the two measurements are within error. Is there a way to decrease the error bars on these measurements? This is a little bit troublesome for a high impact journal like Nature Materials.

Reply 1-1:

We agree with the reviewer that the statistical significance of the lap-shear results is the basis of our study. We received a similar comment from Reviewer #2 (Comment 3) regarding the significance of the adhesive strengths. Although the success rate of the lap shear test is very low due to the brittle nature of the silicon substrate (please see Reply 3 to Reviewer #2), we conducted follow-up experiments to increase the number of tests and achieved a significant difference between the OH-terminated interface (34.8 ± 1.2 , N=8) and H-terminated interface (30.8 ± 1.8 , N=5). We corrected the values as follows.

Pages 6-7, lines 111-113:

Lap-shear tests were conducted to measure the adhesive strengths of the OH- and H-terminated interfaces, yielding values of 34.8 ± 1.2 ~~32.6 ± 2.1~~ and 30.8 ± 1.8 ~~30.9 ± 2.1~~ MPa, respectively.

Comment 1-2:

The enhancement in the N-K/C-K composition near the OH interface (Figure 5a) seems to be larger than

the uncertainty, but only in the first data point 1 nm away from the interface. At 2 nm and beyond, the compositions are within error. The entire notion of enhanced composition seems to be based on this single datapoint. In figure 4c and 4d, the authors show raw spectra of the C-K edge and claim there is qualitative decrease in the π^* peak near the -OH surface that would support enrichment of the PACM, but those peak (or shoulder) heights are not quantified. Why did they not quantify this enhancement? I would really like to see a stronger quantitative basis for this enrichment.

Reply 1-2:

Following the reviewer's comment, we quantitatively evaluated the *stoichiometric ratios* of the epoxy resin near the interfaces by the two approaches, (i) the ratio of the π^* peak/ σ^* peak in each C-K edge and (ii) that of the integrated intensities of the corresponding N-K and C-K edges. As the reviewer mentioned, the compositional change at only one point near the interfaces in the N-K/C-K intensity profiles may not be reliable. However, in the revised manuscript, the profiles of the stoichiometric ratios estimated from the two different approaches exhibited similar trends, which assured the credibility of our results (the composition of the epoxy resin differs near the interfaces). Furthermore, the ATR-FTIR analysis presented in the Supporting Information also supports the result. We revised the manuscript and introduced a new section (S10) into the Supporting Information to explain the procedure to calculate the stoichiometric ratios near the interfaces, along with modifications to the main text as follows.

Pages 14-15, lines 243-254 and Figure 5:

The profiles of the stoichiometric ratio (amine-to-epoxy ratio) were evaluated by dividing the π^* peak intensity by the σ^* peak intensity in the C-K edge spectrum, as described in Section S10 in the Supporting Information. These profiles in Figure 5 (red circles) show that the stoichiometric ratio at the OH-terminated interface (~1.3) is larger than that in the bulk (1.0). In contrast, the epoxy resin close to the H-terminated interface shows a stoichiometric ratio of ~0.7, which is smaller than that of the bulk region.

Figure 5. Profiles of the stoichiometric ratio (amine-to-epoxy ratio) measured at different distances from the (a) OH-terminated and (b) H-terminated adhesive interfaces. The red circles represent the stoichiometric ratio estimated from the intensity of the π^* peak divided by that of the corresponding σ^* peak of the C-K edge spectrum (C-K π^*/σ^*). The blue squares indicate the integrated intensity of the N-K edge spectrum divided by that of the corresponding C-K edge spectrum (N-K/C-K).

Page 15, lines 256-257:

To verify the stoichiometry changes near the interfaces, compositional analysis of the epoxy resin was conducted using N-K-edge ELNES of the OH- and H-terminated interfaces.

Page 15, lines 266-272:

The blue squares in Figures 5(a) and 5(b) depict the stoichiometric ratios calculated from the N-K/C-K intensities plotted against the distance from the OH- and H-terminated interfaces, respectively. In the case of the OH-terminated interface, the stoichiometric ratio of PACM is higher near the oxide-layer interface compared to the bulk region of the epoxy resin. Conversely, a gradual decrease in the stoichiometric ratio is observed for the H-terminated interface from the bulk toward interfacial regions. These profiles are consistent with those obtained from the C-K edge analysis.

Supporting Information, pages 12-13, Figure S9:

Figure S9. Profiles of the integrated intensity of the N-K edge spectrum divided by that of the corresponding C-K edge spectrum (N-K/C-K) measured at different distances from the adhesive interfaces. Profiles of the (a) OH-terminated and (b) H-terminated interfaces.

Supporting Information, pages 13-14, Section S10 and Figure S10:

S10. Estimations of the stoichiometric ratios from STEM-EELS spectra

The stoichiometric ratios of PACM to DGEBA were evaluated from EELS data by using the following two approaches.

(i) N-K peak vs. C-K peak (N-K/C-K)

The integrated intensities of the N-K and C-K edge spectra (I_{N-K} and I_{C-K}) were converted to the ratio the density of the N atoms in PACM (N_N) to that of the C atoms in both DGEBA and PACM (N_C) using a k factor as follows⁵.

$$\frac{N_N}{N_C} = k \cdot \frac{I_{N-K}(\beta, \Delta_{N-K})}{I_{C-K}(\beta, \Delta_{C-K})} \quad (\text{S3})$$

where β is the collection angle of the EELS spectrometer, and Δ is the energy range used to integrate the spectral intensity. In this study, Δ_{C-K} and Δ_{N-K} were set to 280–310 and 390–420 eV for the C-K and N-K edge spectra, respectively. The k factor (a constant value independent of the stoichiometric ratio) was calculated by measuring the intensity ratio (I_{N-K}/I_{C-K}) from the epoxy resin with a stoichiometric ratio of 1.0, where N_N/N_C should be 2/55. Once the k factor was determined, the stoichiometric ratio at each position in the epoxy resin near the interfaces was estimated by dividing the right term in Equation S3 ($k \cdot I_{N-K}/I_{C-K}$) by 2/55. Here, we assumed that the stoichiometric ratio of the epoxy resin in the regions 10–15 nm away from the interface was 1.0 for both OH- and H-terminated specimens.

(ii) C-K π^* peak vs. C-K σ^* peak (C-K π^*/σ^*)

The stoichiometric ratio of PACM to DGEBA was evaluated from the intensity ratio of a C-K π^* peak to σ^* peak. The π^* peak intensities of C-K spectra reflect the amount of the C atoms in the benzene rings of the DGEBA molecules. The intensities of C-K σ^* peaks represent the C atoms not included in the benzene rings. Spectral fitting with three Gaussian functions was performed for the C-K edge spectra as shown in Figure S10. Whereas the first peak (P₁) corresponds to the π^* peak and the second (P₂) and third (P₃) peaks represent the other peaks. The ratio of the P₁ intensity to the P₂+ P₃ intensities is described as follows.

$$\frac{N_{\pi}}{N_{all}} = k \cdot \frac{I_{P_1}}{I_{P_2} + I_{P_3}} \quad (S4)$$

Here, N_{π} is the number of C atoms in the benzene rings in DGEBA, and N_{all} is the total number of C atoms consisting of DGEBA and PACM in the epoxy resin. The k factor was determined in the same way as described in Equation S3. The ratio of N_{π}/N_{all} should be 24/55 for a stoichiometric ratio of 1.0. Therefore, the stoichiometric ratio at each position in the epoxy resin near the interfaces was estimated by dividing the right term in Equation S4 ($k \cdot I_{P_1}/(I_{P_2} + I_{P_3})$) by 24/55.

Figure S10. Decomposition of a C-K absorption spectrum. Three gaussian functions were applied for the spectral fitting.

Comment 2:

2. While I tend to believe there is some form of enrichment of the PACM near the -OH interfaces, I am not entirely sure the chemical mechanisms are that lead to the enhanced mechanical properties. Generally, primary amines (-NH₂) in the PACM should not react with the silanol (Si-OH) groups at the interface. The two groups both too nucleophilic. So, what is the mechanism by which the cross-linking density increases near the interface? Just the enrichment of PACM near the surface alone would not seemingly lead to increased cross-link density. Is this a catalytic effect? It is well known that -OH groups can catalyze the reaction between -NH₂s and epoxides. Is it more the presence of the surface -OH's that drives the reaction to a greater extent near the surface than the excess PACM that is responsible for their observations? Enrichment of PACM near the interface alone would seem to drive the composition off-stoichiometry and actually reduced the cross-link density.

Reply 2:

We appreciate the reviewer's suggestive advice. Please see the response to Comment 5 of Reviewer #2 as this answers similar queries. The manuscript was revised as follows.

Pages 17-20, lines 299-336 and Figure 7:

Curing molecular dynamics simulation on epoxy resin-silica interfaces

Subsequently, curing MD simulations of interfacial models were performed to explore the cross-linked and adhesion structures close to the OH- and H-terminated interfaces in detail. In the simulation, the (10 $\bar{1}$ 0) surfaces of α -quartz were terminated with OH and H as models of the oxidized Si layers, as shown in Figure 7(a). The uncured epoxy resin, consisting of DGEBA and PACM molecules, was confined between the OH- and H-terminated surfaces. The average stoichiometric ratios of the uncured epoxy resins within the OH- and H-terminated systems were set to reflect the interfacial compositions measured by STEM-EELS. Then, curing MD simulations were performed on these systems. Note that the simulations did not consider chemical reactions between the epoxy resin molecules and the substrates.

The stoichiometric ratios of the cured epoxy resins within 1 nm of the OH- and H-terminated interfaces (grey region in Figure S11) are approximately 1.5 and 1.0, respectively, which are similar to the experimentally measured ratios. Figure 7(b) depicts the number density distributions of the reacted amino groups (secondary and tertiary amines) in the epoxy resins near the OH- and H-terminated interfaces. Within 1 nm of the interface (grey regions), the number densities of tertiary amines (branched structures) are almost equal for both the OH- and H-terminated interfaces, whereas the OH-terminated interface, with a higher stoichiometric ratio, has more secondary amines (linear structures) than the H-terminated interface. This tendency is in good agreement with that of bulk systems, as shown in Figure 6(c). In other words, the OH-terminated interface has a higher average molecular weight between the cross-links than the H-terminated interface. The presence of distinct cross-linked structures at the interfaces could result in different mechanical properties.²⁵

The red lines in Figure 7(b) show the distributions of the hydroxyl groups, originating from the reactions between epoxy and amino groups, as shown in Figure S5. The OH-terminated system exhibits a higher average density of the hydroxyl groups in the epoxy resin compared to the H-terminated system. This originates from the higher stoichiometric ratio in the OH-terminated system, as observed for the bulk system (Figure 6(c)). Furthermore, the adsorption layers of hydroxyl groups exhibit a higher density near the OH-terminated interface compared to that near the H-terminated interface (denoted by black triangles). This difference arises from the formation of hydrogen bonds between the hydroxyl groups on the silica surface (silanol groups) and the hydroxyl groups generated within the epoxy resin.

Figure 7(c) illustrates the simulated atomic structure of the OH- and H-terminated interfaces. The hydrogen bonds (hydrogen-oxygen distance: <0.25 nm) between the silanol groups of the substrate and hydroxyl

groups of the epoxy resin are indicated by green lines. The illustration indicates that numerous hydrogen bonds are formed at the OH-terminated interface, while there are very few hydrogen bonds at the H-terminated interface. Consequently, the OH-terminated interface with a PACM-rich composition generates many hydroxyl groups in the epoxy resin and preferentially forms hydrogen bonds to stabilize the adsorption structure.

Figure 7. Epoxy resin structures cured at the OH- and H-terminated interfaces. (a) Simulation model before curing and the surface structures of the OH- and H-terminated SiO₂ (α -quartz) used for the curing molecular dynamics simulations. The top faces of the SiO₂ crystals are terminated with hydroxyl groups and hydrogen atoms, respectively. (b) Number densities of the secondary and tertiary amines and hydroxyl groups in the epoxy resins near the OH- and H-terminated interfaces. The number densities of Si and O atoms in the substrates are shown by black dotted lines to clarify the positions of the interfaces. (c) Schematics of the OH- and H-terminated interfaces.

Page 21, line 356-357:

The E_{ad} values estimated for the OH- and H-terminated interfaces from the curing MD simulations were 0.223 and 0.147 J/m² ~~0.218 and 0.153 J/m²~~, respectively.

Page 25, Figure 9:

Figure 9. Schematics of the interfacial structures and fracture paths near the (a) OH-terminated and (b) H-terminated interfaces.

Comment 3:

3. The mechanical tests need to be explained in a little greater detail. The macroscopic lap shear experiments clearly indicate the interface was loaded in shear. However, the in-situ TEM experiments seem to be loading that interface in tension? The manuscript is not clear about this detail. Were the two loading conditions the same? Or different? If different, how does this play into the interpretation?

Reply 3:

Tensile loads were applied perpendicular to the interfaces for the TEM observation of the fracture surfaces. Thus, the loading conditions for the lap shear test and fracture for the TEM observation are indeed different. First, we note here that tensile adhesive strength data would be preferable for discussing the relationships between the macroscopic adhesive strengths and microscopic fracture surfaces. However, the tensile adhesive strength test is challenging to perform compared to the lap shear test because the test setup and variations in the resin thickness lead to more significant non-uniformity in the stress distribution across the adhesion area. In addition, because there are few previous studies for comparison, the validity of the obtained values is uncertain. On the other hand, there are abundant lap shear data from previous experiments for comparison. Thus, in this study, we developed a technique for lap shear tests that are applicable to the silicon substrates. However, to respond specifically to the reviewer's question, we also developed a tensile adhesive strength test in this revision. As a result, we achieved to obtain the adhesive strengths of 11.4 ± 1.6 MPa (N=7) and 10.4 ± 3.9 MPa (N=8) for the OH- and H-terminated interfaces, respectively. Although the errors for both values are large, the mean values (11.4 and 10.4 MPa) suggest the same strength

relationship as those measured by the lap shear test. In the revised manuscript, we added further descriptions as follows.

Page 22, lines 377-382:

Subsequently, tensile loads were applied perpendicular to the interfaces using a tensile TEM holder to induce fracture, as depicted in Figures 8(a) and (b).²⁷⁻²⁹ Although the loading conditions applied to the specimens of the lap-shear test were different from those of the tensile fracture tests with the TEM holder, we presumed that the strength relationships between the OH- and H-terminated interfaces were similar for both tensile and shear deformation since interfaces with weaker bonds are more likely to break at lower loads.

Supporting Information, pages 16-17, lines 270-285 and Figure S13:

Figure S13 shows schematics of the tensile adhesive strength test setup. In the test, two OH- and H-terminated Si substrates, each measuring 25 mm in width and 25 mm in length, were bonded with the DGEBA–PACM epoxy resin. A single-sided release film with a rectangular hole (25 mm width and 5 mm length) filled with a mixture of DGEBA and PACM was sandwiched between the two Si substrates. The assembly was then heated at 100 °C for 90 min to cure the epoxy resin. The region of the epoxy resin measured 25 mm in width, 5 mm in length, and 0.19 mm in thickness.

Furthermore, SUS304 blocks, measuring 25 mm in width, 25 mm in length, and 12 mm in thickness, were bonded to both sides of the test pieces using a room-temperature curing adhesive. The test pieces were mounted on a universal testing machine (Instron 68FM-300, Instron Corporation, USA) with a 250 kN static load cell. The tensile adhesive strength tests were conducted at a tensile speed of 0.02 mm/min. The adhesive strength was calculated by dividing the load-at-break by the adhesion area of 25 mm × 5 mm.

As a result, we obtained the adhesive strengths of 11.4 ± 1.6 and 10.4 ± 3.9 MPa for the OH- and H-terminated interfaces, respectively. Although the errors for both values are large, the mean values (11.4 and 10.4 MPa) imply a similar strength relationship as those measured by the lap shear test.

Figure S13. Schematics and photographs of the tensile adhesive strength test setup.

Comment 4:

4. The TEM fracture data presented in Figure 8 is used to argue that there is more cohesive failure in the -OH samples while the -H samples exhibit more interfacial delamination. Clarifications to point 3 above will help interpret these results. I would agree that there does seem to be more mechanical drawing of the resin near the -OH substrate, and more clean fracture near the -H substrate, but I need to know more details about the test. In bulk DGEBA / PACM resin, in the process zone of a crack near the initiation site, where the crack is slowly growing and damage is building up, you can see local drawing of the material. However, once the crack reaches its critical length and starts to propagate catastrophically, the fast fracture surfaces look smooth. This could also be an explanation for the difference between Figures 8c and 8d. I need to know more about where these images came from with regards to the process zone and fast fracture zone to have confidence in the authors' assertion.

Reply 4:

To investigate the crack propagation zones, we performed FEM simulations on a model of the interface specimens for TEM observation. The conditions and results of this simulation are described in a new section (S13) and Figs. S14-S16 in the Supporting Information. In this simulation, a pre-crack (length: 10 nm) was made at the ends of the interface, assuming that the crack initiates from the ends of the interface due to a stress concentration under tensile loading. We assumed that the fracture in the specimen occurred at 9.8 MPa because the measured macroscopic tensile adhesive strengths of the OH- and H-terminated interfaces were ~10 MPa. However, since 9.8 MPa is still in the elastic deformation region, no process zone or HRR singular field was observed in the TEM specimen. Therefore, the crack initiates and then propagates once the stress intensity factor at the crack tip reaches the fracture toughness value of the interface. In other words, the crack propagates rapidly outside the pre-cracked region. Therefore, we conclude that the entire interfacial region shown in Figure 8 was included in the fast fracture zone. We added the following section in the Supporting Information to explain the details of the FEM analysis.

Supporting Information, pages 17-20, lines 287-328 and Figures S14-S16:

S13. Stress-field analysis around a crack tip at the epoxy resin-silicon substrate interfaces

A finite element model (FEM) of the interfacial specimens used for the TEM observation of the fracture surfaces was created in the explicit finite element code LS-Dyna. As shown in Figure S14, the shapes and dimensions of the model were set to replicate the specimen shown in Figure 8(b). The material properties of the Si substrate and epoxy resin were defined by isotropic elastic and isotropic elastoplastic models, respectively. The material models of *MAT_001 and *MAT_012 were used for the simulation in the LS-Dyna software. The equivalent stress and work hardening rate, which were input into *MAT_012, were determined based on the stress-strain curves obtained from uniaxial tensile tests and FEM analysis of the epoxy resin. The model was composed of the epoxy resin and silicon parts with C3D8 elements across the

300 nm thickness. The model had 94,772 nodes and 70,152 elements. The model was divided into three sections of 100 nm in the thickness direction. The widths and lengths of the divisions were set in the range of 10–200 nm and 10–500 nm, respectively. The elements at the edges of the epoxy resin–silicon interface were divided into 10 nm × 10 nm × 100 nm volumes.

Assuming that crack formation occurs at the ends of the interface due to the stress concentration involved in the Poisson shrinkage, pre-cracks were introduced in the elements at both ends of the interface. No fracture was considered for the other parts of the interface. To reproduce the experimental setup (Figure 8b), while the displacement of the hatched area on the silicon substrate (representing the nodes on the silicon surface) in Figure S14 was constrained, the nodes in the hatched area on the epoxy resin were subjected to displacement in the y direction. The macroscopic tensile adhesive-strength tests of the epoxy resin–Si substrate interfaces (where tensile loads were applied perpendicularly to the interfaces) exhibited delamination at ~10 MPa. Therefore, the stress distribution at the pre-crack tip was analysed with a simulated tensile stress of 9.8 MPa, assuming that the delamination occurred at the similar stress in the interfacial specimens used for TEM observations.

Figure S14. Schematic illustration of the finite element model and mesh at the epoxy resin–silicon substrate interface.

Figure 15 shows the stress–strain curve obtained from the FEM analysis. As mentioned above, this analysis did not model the fracture or interfacial delamination between the epoxy resin and silicon substrate. The tensile loading was applied until the epoxy resin plastically deformed. The experimentally obtained adhesive strength (~10 MPa) is in the elastic-deformation region where the stress is linearly increasing. Figure 16 shows the stress distribution near the interface at ~10 MPa and variation in the tensile stress σ_y of the epoxy resin elements close to the interface, which is plotted as a double logarithmic graph against the distance r along the interface from the crack tip. Because we assumed that the cracks were generated at the epoxy resin–silicon interface, although the relationship between σ_y and r did not show the singularity of $1/\sqrt{r}$, there was no process zone and HRR singular field. Therefore, the crack starts to propagate once

the stress intensity factor at the crack tip reaches the fracture toughness value of the interface. In other words, the area outside the pre-crack is the region where the crack propagates rapidly during the final fracture stage. In this case, the crack propagates along with the fracture occurring at regions with weak bonds located in the direction of crack propagation. If the adhesive strength of the epoxy resin–silicon interface is less than the cohesive strength of the epoxy resin, the crack propagates along the interface, and vice versa.

Figure S15. Simulated stress–strain curve of the epoxy resin–Si interfacial model.

Figure S16. (a) Distribution of tensile stress (σ_y) around the interfacial crack tip and (b) σ_y distribution along the interface between the epoxy resin and Si substrate.

REVIEWERS' COMMENTS

Reviewer #2 (Remarks to the Author):

The changes made to the manuscript have addressed the reviewer comments satisfactorily, and the manuscript is much improved. I recommend publication, my only suggested amendment would be further work up of the abstract and conclusions to highlight the novelty of the work. It should be noted that this paper represents the first clear, direct experimental evidence that molecular surface segregation and cross-linking in such systems can be driven by intermolecular forces at the surface alone, and indeed that this can be linked directly to adhesive failure mechanisms.

Reviewer #3 (Remarks to the Author):

I would like to thank the authors for their complete and thorough consideration of my review comments (Reviewer #3). After reviewing their responses to my comments, and resulting changes to the manuscript, my concerns are alleviated. I also see that Reviewer #2 had similar concerns to my own. This is reassuring. Now that the revisions have been made, I now support publication.

Please note that modifications are highlighted in **red** in the revised manuscript.

Reply to Reviewer #2

The changes made to the manuscript have addressed the reviewer comments satisfactorily, and the manuscript is much improved. I recommend publication, my only suggested amendment would be further work up of the abstract and conclusions to highlight the novelty of the work. It should be noted that this paper represents the first clear, direct experimental evidence that molecular surface segregation and cross-linking in such systems can be driven by intermolecular forces at the surface alone, and indeed that this can be linked directly to adhesive failure mechanisms.

Reply:

We revised the manuscript according to the reviewer's advice to highlight the novelty of our study as follows.

Page 3 (Abstract):

The mechanisms underlying the influence of the surface chemistry of inorganic materials on polymer structures and fracture behaviours near adhesive interfaces are not fully understood. **This study demonstrates the first clear and direct evidence that molecular surface segregation and cross-linking of epoxy resin are driven by intermolecular forces at the inorganic surfaces alone, which can be linked directly to adhesive failure mechanisms.**; ~~this was studied herein using electron microscopy and molecular dynamics simulations at the molecular scale for the first time.~~ We prepare adhesive interfaces between epoxy resin and silicon substrates with varying surface chemistries (OH and H terminations) with a smoothness below 1 nm, with different adhesive strengths by ~13 %. The epoxy resins within sub-nanometre distance from the surfaces with different chemistries exhibit distinct amine-to-epoxy ratios, cross-linked network structures, and adhesion energies. The OH- and H-terminated interfaces exhibit cohesive failure and interfacial delamination, respectively. The substrate surface chemistry impact the cross-linked structures of the epoxy resins within several nanometres of the interfaces and the adsorption structures of molecules at the interfaces, which result in different fracture behaviours and adhesive strengths.

Pages 25-27 (Conclusions):

This study **first** examined the effects of the surface chemistry (OH and H termination) of Si substrates on the composition, cross-linked structure, and fracture mechanisms of epoxy resin near adhesive interfaces. Lap shear tests were conducted to quantify the adhesive strength, showing that the OH-terminated interface exhibited ~13% higher adhesive strength than the H-terminated interface. To understand the underlying reasons for this discrepancy in adhesive strength, high-resolution TEM/STEM and curing MD simulations were employed for detailed analysis.

The Si-L_{2,3} edge ELNES spectra provided insights into the H termination remaining even after the resin was applied and cured on the surface. In contrast, the C-K and N-K edge ELNES spectra indicated that the PACM curing agent condensed (resulting in a higher stoichiometric ratio than that of the bulk) within a 1–2 nm range at the OH-terminated interface, whereas the opposite phenomenon was observed at the H-terminated interface (smaller stoichiometric ratio than that of the bulk). These findings were corroborated by ATR-FTIR measurements. The Si substrates have a sufficiently flat surface with a roughness of ~0.5 nm, and the oxide layers have comparable thicknesses at both the OH- and H-terminated interfaces. Given these conditions, we conclude that the disparity in surface chemistry solely accounts for the variations in chemical composition near the interfaces.

Curing MD simulations further indicated that the compositional differences in the epoxy derived from the surface chemistry of the Si substrate influenced the cross-linked structures near the interfaces. Additionally, the adhesion energy at the OH-terminated interface was higher (indicating greater stability) than that of the H-terminated interface due to the adsorption of the epoxy resin on the OH-terminated surface via hydrogen bonds.

TEM observations of the fractured surfaces revealed distinct fracture behaviours between the OH-terminated surface and H-terminated interfaces. The OH-terminated interface exhibited cohesive failure due to its higher adhesion energy, while the H-terminated interface showed partial interfacial delamination due to its lower adhesion energy.

These findings clearly demonstrated the significant influence of the intermolecular interactions at adhesive interfaces ~~the surface chemistry of inorganic substrates on both the molecular segregation and cross-linked structures of epoxy resin interfacial interactions between the epoxy resin and substrates and the epoxy resin cross-linked structures~~ near the interfaces at a single-nanometre scale. The structural changes in the epoxy resin due to the surface chemistry near the interfaces are directly linked to ~~have implications for~~ the interfacial adhesive strengths and the propagation path of cracks near the interfaces. The acquired knowledge of the chemical interactions at the adhesive interfaces contributes to a comprehensive understanding of the interfacial adhesion and fracture mechanisms between polymers and inorganic materials.

Reply to Reviewer #3

I would like to thank the authors for their complete and thorough consideration of my review comments (Reviewer #3). After reviewing their responses to my comments, and resulting changes to the manuscript, my concerns are alleviated. I also see that Reviewer #2 had similar concerns to my own. This is reassuring. Now that the revisions have been made, I now support publication.

Reply:

We appreciate reviewer #3 for his/her valuable and insightful comments, which substantially improved our manuscript. We are glad that the reviewers thought our revisions were satisfactory.